# Performance of the Adriatic early warning system during the multi-meteotsunami event of 11-19 May 2020: an assessment using energy banners

Iva Tojčić[1*], Cléa Denamiel[1,2], Ivica Vilibić[1]

[1] Institute of Oceanography and Fisheries, Šetalište I. Meštrovića 63, 21000 Split, Croatia
[2] Ruđer Bošković Institute, Division for Marine and Environmental Research, Bijenička cesta 54, 10000 Zagreb, Croatia

*Correspondence to*: I. Tojčić, (tojcic@izor.hr)

**Abstract.** This study quantifies the performance of the Croatian meteotsunami early warning system (CMeEWS) composed of a network of air pressure and sea-level observations, a high-resolution atmosphere-ocean modelling suite and a stochastic surrogate model. The CMeEWS, which is not operational due to a lack of numerical resources, is used retroactively to reproduce the multiple events observed in the eastern Adriatic between the 11[th] and 19[th] of May 2020. The performances of the CMeEWS deterministic models are then assessed with an innovative method using energy banners based on temporal and spatial spectral analysis of the high-pass filtered air pressure and sea-level fields. It is found that deterministic simulations largely fail to forecast these extreme events at endangered locations along the Croatian cost mostly due to a systematic north-westward shift of the atmospheric disturbances. Additionally, the use of combined ocean and atmospheric model results, instead of atmospheric model results only, is not found to improve the selection of the transects used to extract the atmospheric parameters feeding the stochastic meteotsunami surrogate model. Finally, in operational mode, the stochastic surrogate model would have triggered the warnings for most of the observed events, but also setting off some false alarms. Due to the uncertainties associated with operational modelling of meteotsunamigenic disturbances, the stochastic approach has thus proven to overcome the failures of the deterministic forecasts and should be further developed.

# 1 Introduction

Atmospherically-driven extreme sea-levels (e.g. wind storms, hurricanes), associated with flooding producing substantial damages to houses, goods, and infrastructures, are among the main hazards impacting the coastal communities (Nicholls and Cazenave, 2010; Neumann et al., al., 2015). As such, meteorological tsunamis (commonly referred as meteotsunamis) are sea-level oscillations with characteristics similar to seismic or landslide tsunamis but generated by atmospheric gravity waves, frontal passages, pressure jumps, squalls, etc., though a multi-resonant mechanism (Monserrat et al., 2006). The principal generation mechanisms are the open-ocean resonance occurring between the ocean and the air pressure oscillations at time scales ranging from a few minutes to a few hours (e.g. Proudman, 1929), as well as the coastal amplification that includes also so-called harbour resonance (Miles and Munk, 1961; Rabinovich, 2009). Locally they can be destructive, not only due to extreme sea-levels (Hibiya and Kajiura, 1982; Salaree et al., 2018), but also to dangerous currents in constrictions or in coastal zone (Ewing et al., 1954; Vilibić et al., 2004; Linares et al., 2019). The strongest meteotsunami on record in the Mediterranean Sea hit Vela Luka, Croatia, in June 1978, with a wave height of 6 m (crest-to-trough) and a period of 18 min. The meteotsunami lasted several hours and caused 7 million US dollar damages (Vučetić et al., 2009; Orlić et al., 2010).

In certain locations around the world, due to a combination of weather patterns, geography and bathymetry, meteotsunamis can be a regularly occurring phenomenon. The Balearic Islands and Croatian coastline in the Mediterranean Sea, a few od Japan's gulfs and bays, the Great Lakes and the US East Coast, western Australian coastline are good examples (Pattiaratchi and Wijeratne, 2015; Rabinovich, 2020). For all these locations despite varying intensities, meteotsunami events have the potential to generate structural damages and sometimes even human casualties. Meteotsunami early warning systems, helping the local population to prepare for these destructive events, are thus important for the coastal communities living in such places. Vilibić et al. (2016) pointed out that meteotsunami early warning systems can be created based on the four approaches: (1) identification of tsunamigenic atmospheric synoptic conditions; (2) real-time detection of tsunamigenic atmospheric disturbances using a microbarograph network; (3) measurement and tracking of high-frequency sea-level oscillations by high-resolution digital tide gauges; and (4) numerical simulation of meteotsunamis based on coupling of atmosphere-ocean numerical models. As it stands today, the only fully operational meteotsunami early warning system in the world is located in the Balearic Islands. It is based on forecasts given both at a qualitative level with the identification of favourable synoptic conditions a few days ahead (Jansà et al., 2007; Jansà and Ramis, 2020) and with the deterministic results of the operational BRIFS (Balearic Rissaga Forecasting System, www.socib.eu) model (Renault et al., 2011). In the Balearic Islands, probabilistic approaches have also been tested recently to narrow down the uncertainties of the meteotsunami forecasts (Vich and Romero, 2020; Mourre et al., 2020). In the United States, meteotsunami early warning systems are still under development by the NOAA (National Oceanic and Atmospheric Administration) and will be based on high-resolution air pressure measurements combined with forecast models (Anderson et al., 2020). Finally, the recently developed Croatian Meteotsunami Early Warning System (CMeEWS) is based on an observational network of pressure sensors and tide gauges, as well as on the deterministic AdriSC modelling suite (Denamiel et al., 2019a) and the stochastic meteotsunami surrogate model (Denamiel et

al., 2019b, 2020). It provides meteotsunami hazard assessments depending on forecasted and measured air pressure disturbances but is, unfortunately, not used operationally since November 2019 due to a lack of high-performance computing resources needed to execute in real-time such numerically demanding suite.

However, the CMeEWS applications to recent meteotsunami events may surely be used to better quantify its reliability and to improve its performance. Recently, an exceptional multi-meteotsunami event, that lasted for a week between the 11[th] and 19[th] of May 2020, occurred in the Croatian cities of Vela Luka (VL), Stari Grad (SG) and Vrboska (Vr), located along the coasts of the Dalmatian islands in the Adriatic Sea (Fig. 1). Therefore, the deterministic and stochastic AdriSC models have been run retroactively in operational (hindcast) mode (i.e. in the exact same conditions than the daily meteotsunami forecasts would have been produced operationally) for this 11-19 May 2020 period. As quoted by Denamiel et al. (2020), forecasting the right speed and frequency (period) of the travelling atmospheric disturbances is crucial for meteotsunami hazard assessments in the harbours of Vela Luka, Stari Grad and Vrboska. Therefore, unlike previous studies on the performances of the CMeEWS operational models, this analysis introduces the novelty of using energy banners – based on the spectral analysis of the high-pass filtered air pressure and sea-level fields – as a tool to evaluate the capacity of the AdriSC deterministic model to reproduce the frequency of the meteotsunamigenic disturbances measured during the 11-19 May 2020 period. Hereafter, the CMeEWS (including the AdriSC modelling suite, the stochastic surrogate model and the observational network) and the methods used in the study are first presented in Section 2. Then, Section 3 describes the 11-19 May 2020 multi meteotsunami event, using eyewitness reports, available observations and reanalysis products. The verification of the AdriSC deterministic atmospheric model is undertaken in Section 4, while Section 5 presents the main results of the study – i.e. the meteotsunami energy banners used to detect the strongest atmospheric disturbances. Finally, the stochastic meteotsunami hazard assessments, based on parameters extracted from transects selected along the energy banners, are discussed in Section 6 and the findings of this study are summarized in Section 7.

## 2 Model, Data, and Methods

The Croatian Meteotsunami Early Warning System (CMeWS) receives three different kind of data: (1) high-resolution atmospheric and ocean model results provided by the Adriatic Sea and Coast (AdriSC) modelling suite (Denamiel et al., 2019a), (2) high-frequency air pressure and sea-level measurements along the Adriatic coast and (3) meteotsunami hazard assessments based on the stochastically estimated maximum elevation distributions derived from a meteotsunami surrogate model (Denamiel et al., 2019b, 2020). The following sub-sections describe the different components of the CMeWS as well as the methods used in this article to improve the detection and extraction of the modelled meteotsunamigenic disturbances in the atmosphere.

## 2.1 AdriSC modelling suite

The AdriSC modelling suite is composed of a basic module providing kilometre-scale atmospheric and ocean circulation over the entire Adriatic region, forcing a dedicated meteotsunami module (Denamiel et al., 2019a).

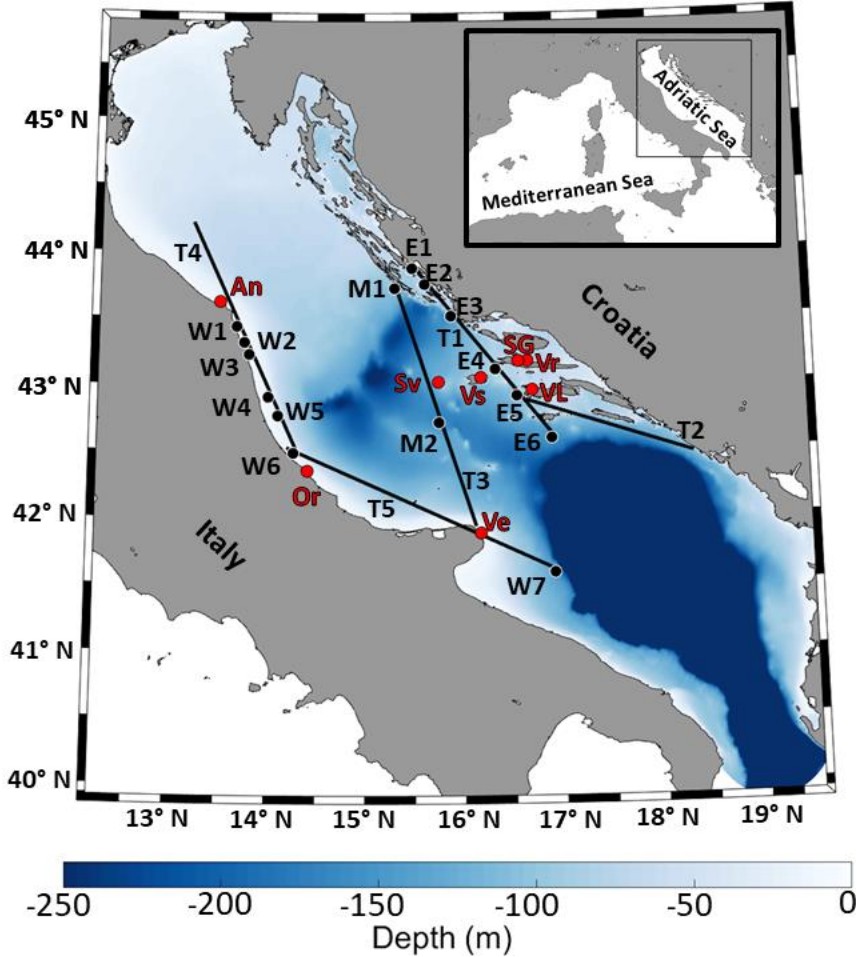

**Figure 1: Bathymetry of the Adriatic Sea with positions of microbarographs and tide gauges (red circles). Black circles denote model grid points along transects T1 to T5 (black lines) on which the highest energy is reproduced within selected meteotsunami energy**
**banners in the eastern (E1 to E6), middle (M1 and M2) and western (W1 to W7) Adriatic.**

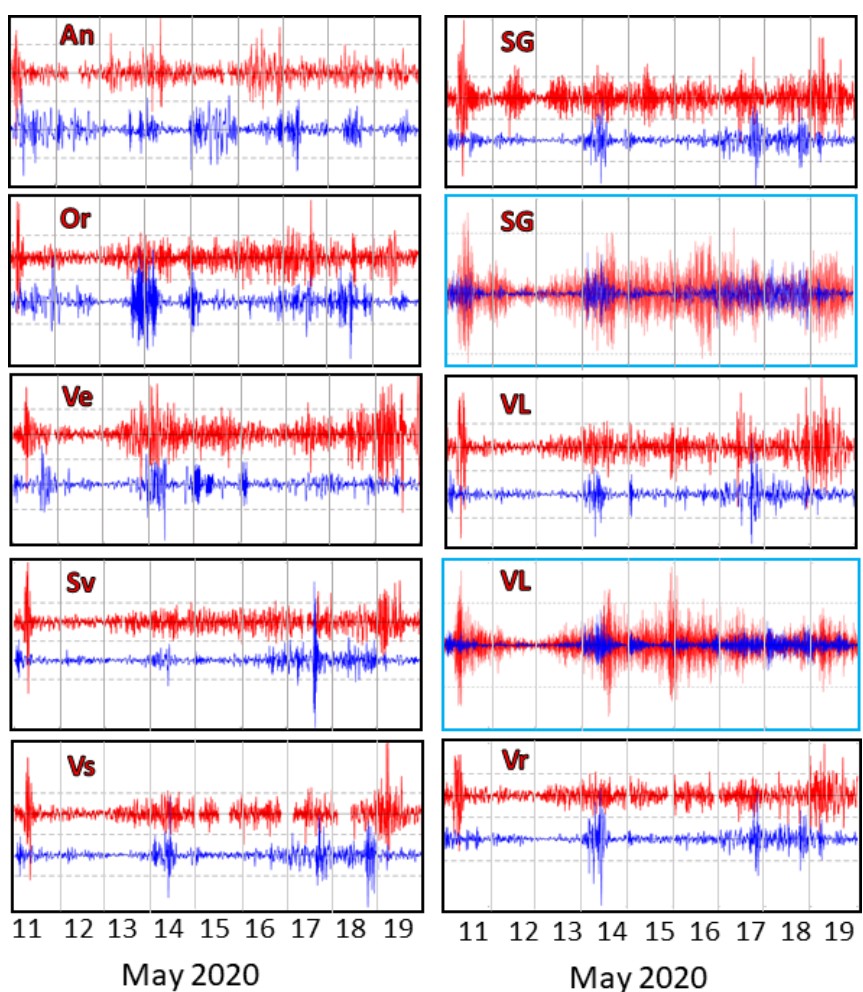

**Figure 2: Observed (in red) and modelled (in blue) high-pass filtered time series of air pressure (black rectangles) and sea-level (blue rectangles) during the 11-19 May 2020 period. Distance between adjacent horizontal grid lines (dashed) stands for 1.0 hPa in air pressure, as well as 0.5 m in sea-level for Stari Grad and Vela Luka.**

The basic module uses a modified version of the Coupled Ocean-Atmosphere-Wave-Sediment-Transport (COAWST) modelling system developed by Warner et al. (2010), built around the Model Coupling Toolkit (MCT) which exchanges data fields and dynamically couples the Weather Research and Forecasting (WRF) atmospheric model, the Regional Ocean Modeling System (ROMS), and the Simulating WAves Nearshore (SWAN) model. The basic module is set-up with (1) two different nested grids of 15-km and 3-km resolution used in the WRF model and covering respectively the central
Mediterranean area and the Adriatic-Ionian region and (2) two different nested grids of 3-km and 1-km resolution used for both ROMS and SWAN models and covering respectively the Adriatic-Ionian region (similarly to the WRF 3-km grid) and the Adriatic Sea only.

The dedicated meteotsunami module couples offline the Weather Research and Forecasting (WRF) model (Skamarock et al., 2005) at 1.5-km of resolution with the unstructured ADCIRC-SWAN model (Dietrich et al., 2012) coupling the 2DDI (i.e. two dimensional depth-integrated) ADvanced CIRCulation (ADCIRC) model and the SWAN model with a mesh of up to 10-m resolution in the areas sensitive to meteotsunami hazard. In more details, (1) the hourly results from the WRF 3-km grid obtained with the basic module are first downscaled to a WRF 1.5-km grid covering the Adriatic Sea and (2) the hourly sea surface elevation from the ROMS 1-km grid, the 10-min spectral wave results from the SWAN 1-km grid and finally the 1-min results from the WRF 1.5-km grid are then used to force the unstructured mesh of the ADCIRC-SWAN model. In this operational configuration, the ADCIRC model is forced every minute by the WRF 1.5-km wind and pressure fields, and every hour by the basic module sea-level fields (including tides) at the open sea boundary (south of the Otranto Strait).

In operational mode (Denamiel et al., 2019a, 2019b) the AdriSC modelling suite runs every day with the basic module initial state and boundary conditions provided by (1) the previous day 12 UTC based analysis of the ECMWF 10-day forecast model (HRES at 0.1° resolution; Zsótér et al., 2014) for the atmosphere and (2) the Mediterranean Forecasting System (MFS/MEDSEA at 1/24° resolution; Pinardi et al., 2003) for the ocean.

## 2.2 Observational network

The observational network (called MESSI, www.izor.hr/messi) consists of nine microbarographs, of which eight are used in this study, measuring air pressure by the Väisälä PTB330 sensor with an accuracy of ±0.01 hPa, and three tide gauges, of which two are used in this study, measuring sea-level by the OTT RLS radar level sensor with an accuracy of ±1 mm. All instruments are setup with a 1-min sampling rate and listed in Table 1. Microbarographs are installed in areas where either the generation or the amplification of meteotsunamis are known to occur (red circles, Fig. 1): Ancona (An), Ortona (Or), and Vieste (Ve) located along the western Adriatic coast, Vis (Vs) and Svetac (Sv) in the middle of the Adriatic Sea, and Vela Luka (VL), Stari Grad (SG) and Vrboska (Vr) on the eastern Adriatic coast. Tide gauges are located in Vela Luka (VL) and Stari Grad (SG), which are known to be harbours sensitive to meteotsunamis (red circles, Fig. 1). However, one should be aware that the tide gauges are located not at the tops of the bays that are normally most affected by meteotsunamis, but about 2 km from the tops, thus the observed high-frequency sea-level oscillations at tide gauges are 2 to 3 times lower than reported by eyewitnesses at the bays' tops.

## 2.3 Stochastic surrogate model

Uncertainties linked to the deterministic forecast of the location, direction, amplitude, speed, period and width of the atmospheric disturbances driving meteotsunami events in the Adriatic Sea are known to be quite large (Belušić et al.., 2007; Šepić et al., 2009; Denamiel et al., 2019a). In other words, it is unlikely for atmospheric deterministic models to forecast meteotsunamigenic disturbances with proper speed and period and at the right location. Consequently, deterministic ocean models often fail to reproduce or underestimate the meteotsunami events in sensitive harbours (e.g. Vela Luka, Stari Grad and Vrboska). In order to improve the meteotsunami hazard assessments in the Adriatic, the meteotsunami stochastic surrogate

model, used to propagate the uncertainties of the atmospheric disturbance parameters extracted from the WRF 1.5-km model to the maximum amplitudes of the meteotsunami waves, was developed within the CMeEWS (Denamiel et al., 2019b, 2020). This model is optimizing a great number of ADCIRC simulations via a generalized Polynomial Chaos expansion (gPCE) method (Xiu and Karniadakis, 2002; Soize and Ghanem, 2004), where a particular simulation is forced by synthetic air pressure fields depending on six stochastic parameters: start location ($y_0$), direction ($\theta$), speed ($c$), period ($T$), amplitude ($PA$), and width ($d$) of the disturbance (Denamiel et al., 2018). These six parameters are assumed to have uniform distributions and are adapted to the middle Adriatic meteotsunamis on the following intervals: $y_0 \in [41.25°$ N, $43.65°$ N$]$, $\theta \in [-\pi/3, \pi/2]$, $c \in [15$ m s$^{-1}$, 40 m s$^{-1}]$, $T \in [300$ s, 1800 s$]$, $PA \in [0.5$ hPa, 4 hPa$]$, and $d \in [30$ km, 150 km$]$.

**Table 1. Microbarograph and tide gauge locations.**

| Location | Coordinates | Area | Observations |
|---|---|---|---|
| Ancona (An) | 13.506 °E 43.625 °N | Western Adriatic | air pressure |
| Ortona (Or) | 14.415 °E 42.356 °N | Western Adriatic | air pressure |
| Vieste (Ve) | 16.177 °E 41.888 °N | Western Adriatic | air pressure |
| Svetac (Sv) | 15.757 °E 43.024 °N | Middle Adriatic | air pressure |
| Vis (Vs) | 16.192 °E 43.057 °N | Middle Adriatic | air pressure |
| Stari Grad (SG) | 16.576 °E 43.180 °N | Eastern Adriatic | air pressure & sea-level |
| Vela Luka (VL) | 16.718 °E 42.962 °N | Eastern Adriatic | air pressure & sea-level |
| Vrboska (Vr) | 16.672 °E 43.181 °N | Eastern Adriatic | air pressure |

Within the CMeEWS, the ranges of the stochastic parameters used as input to the surrogate model are extracted manually from the forecasted WRF 1.5-km high-pass filtered air pressure results, adding the uncertainty of ±0.24° N for latitude of origin, ±0.26 rad for direction of propagation, ±0.35 hPa for amplitude, ±150 s for period, and ±12 km for width, following the values determined by Denamiel et al. (2019b). For each sensitive location along the Croatian coast, the output of the surrogate model consists in the distribution of maximum elevations produced with 20 000 random combinations of the input parameters selected within the defined ranges. Additionally, to provide a meteotsunami hazard assessment derived from the surrogate model, Denamiel et al. (2019b) prescribed a flooding threshold – defined as the maximum elevation above which flooding would occur – considering the resilience of the coastline at the different sensitive locations. For Vela Luka, Stari Grad and Vrboska, these thresholds are defined as 1.05 m, 0.45 m and 0.55 m respectively. In operational mode, the meteotsunami warning is

triggered when the probability of crossing the flooding threshold (derived from the maximum elevation distributions provided
as the surrogate model output) is above or equal to 10 %.

## 2.4 Methods

In order to evaluate the capacity of the CMeEWS to provide meaningful meteotsunami hazard assessments, the AdriSC
modelling suite is run in operational (hindcast) mode after the 11-19 May 2020 multi-meteotsunami event took place. This
means that the 10-day forecasts derived with the ECMWF HRES and MEDSEA/MSF models on the 8[th], 9[th], 10[th], …, 16[th] of
160 May 2020 are used to hindcast the meteotsunamigenic conditions of the 11[th], 12[th], 13[th], …, 19[th] of May 2020. The model is
set-up to run for short periods of three days in the basic module and one and a half day in the extreme event module, with only
the last 24-h hourly results – extracted from the WRF 1.5-km model in the atmosphere and the ADCIRC unstructured model
in the ocean – used in the following analyses. Within the CMeEWS, the meteotsunamigenic disturbances reproduced with the
AdriSC WRF 1.5-km model are automatically detected if the maximum temporal rate of change (i.e. pressure difference
calculated over a 4-min interval) of the high-pass filtered air pressure derived at each WRF 1.5-km grid sea point is above 20
Pa/min over at least 5% of the sea domain. Such a condition has been proven to be efficient for the detection of
meteotsunamigenic disturbances (Vilibić et al., 2016; Denamiel et al., 2019b). The event mode of the system (i.e.
meteotsunamis may occur) is thus triggered without human intervention for the studied 11-19 May 2020 period.
Hereafter, air pressure and sea-level data both derived with the AdriSC modelling suite and collected from the stations listed
in Table 1, are filtered using a 2-h Kaiser–Bessel filter to extract high frequency pressure and sea-level oscillations
characteristic for meteotsunamis. At a very basic level, a direct comparison of modelled (blue lines, Fig. 2) and measured (red
lines, Fig. 2) high-pass filtered air pressure and sea-level time series is used in Section 3 to assess the capacity of the AdriSC
deterministic model to reproduce the meteotsunami events at the locations of interest during the middle Adriatic multi-
meteotsunami event of 11-19 May 2020.
Since the failure of deterministic models to reproduce the small-scale atmospheric disturbances at the right locations is a known
problem, the verification of the AdriSC WRF 1.5-km results presented in Section 4 tracks the locations where the highest daily
spectral energies occur in both the model and the observations. In other words, the performance of the AdriSC WRF 1.5-km
model is derived with Fast Fourier Transforms (FFT) analyses (Cooley and Tukey, 1965) of the high-pass filtered air pressure
observed and modelled results calculated every 30 min with a 3-h window at selected locations for each day of the reproduced
multi-meteotsunami event. First, as the meteotsunamigenic disturbances are known to propagate from the Western to the
Eastern Adriatic (Vilibić and Šepić, 2009; Denamiel et al., 2020), 5 transects are selected to track the modelled atmospheric
disturbances: 2 transects along the Italian coast in the Western Adriatic (T4 and T5), one in the Middle Adriatic (T3) and two
transects along the Croatian coast in the Eastern Adriatic (T1 and T2). Then, for each day of the multi-meteotsunami event,
the AdriSC WRF 1.5-km results are extracted at the actual microbarograph locations and in additional model grid points (black
dots, Fig.1) selected where the highest daily spectral energies are reproduced by the model along the Western (selected points

W1 to W7), Middle (selected points M1 and M2), and Eastern Adriatic (selected points E1 to E6) transects. The measurements at the microbarograph location where the meteotsunami was best observed – i.e. highest spectral energy along the Western Adriatic transect for Ancona, Ortona and Vieste microbarographs, along the Middle Adriatic transect for Vis and Svetac microbarographs and along the Eastern Adriatic transect for Vrboska, Stari Grad and Vela Luka microbarographs - are also extracted. Finally, the time evolutions of the spectra derived from the observations (at the selected stations) are compared with the time evolutions of the spectra derived from the WRF 1.5-km results at the point where the highest energy was reproduced (including microbarograph locations). At the end, for the entire duration of the multi-meteotsunami event, composites of frequency-time spectrograms of high-pass filtered air pressure observed and modelled data for the Western, Middle and Eastern Adriatic regions are created (Figs. 4-6).

The analyses performed in Section 5 are done in two steps and aim to better track the propagation of the modelled meteotsunamigenic disturbances across the Adriatic Sea in order to improve the extraction of the atmospheric parameters needed to run the stochastic surrogate model. In the first step, two different transect sampling criteria are used to select the transects along which the atmospheric disturbances, and hence the meteotsunami waves, propagate in the model: one based solely on the atmospheric results (already used operationally) and a new one also taking into account the ocean results (tested in this study). For the operational sampling criterion, the time variances of the WRF 1.5-km high-pass filtered air pressure results are calculated on a 3-hour interval (i.e. 8 time-windows per day) over the entire model domain. For each event occurring during the 11-19 May 2020 period, the transects presented in this study are manually selected across the Adriatic Sea following the paths of highest atmospheric variances for the most energetic time-windows. Since the number of time-windows and paths with high air pressure variances varies between the events, the number of transects for each day varies too. For the new sampling criterion, the variances of the high-pass filtered air pressure and sea-level model results estimated on a 3-hour interval are multiplied. This criterion thus tends to zero when the atmospheric forcing does not trigger any ocean response, i.e. when no resonant transfer of energy from the atmosphere to the sea is occurring. It should be noted that such a criterion could not be directly derived from the sea-level variances which provide a noisy and mostly untraceable signal due to the numerous interactions of the ocean waves with the bathymetry including, for example, the reflection and refraction around the islands. Hereafter the new transect sampling criterion is compared with the operational one in order to determine whether or not it would have improved the transect selection. In the second step, meteotsunami energy banners defined as the spectrograms of the modelled high-pass filtered air pressure and sea-level results are spatially calculated with FFT along the selected transects for the 3-h time-window corresponding to the operational transect sampling criterion. As speed remains a difficult parameter to extract from the observed and modelled meteotsunamigenic disturbances, speeds of the tracked atmospheric disturbances along the transects are also visually determined by analysing the propagation along the transects of the strongest WRF 1.5-km high-pass filtered air pressure peaks. The locations where the Proudman resonance is likely to occur along the transects are then derived by calculating where the Froude number ($Fr=U/C$) ranges from 0.9 and 1.1 (i.e. where the speed of the atmospheric disturbances $U$ are matching the speed of the long ocean waves $C=\sqrt{gH}$, with $g$ the gravitational acceleration and $H$ the local depth). The analyses from Section 5 are presented with one transect (plotted from West to East following the

propagation of the meteotsunami events) per event in the article (Transect 1, Figs. 7-11) selected during the peak of the modelled daily event and as supplementary material for the other transects (Figs. S2-S15) in order to keep a reasonable article length.

Finally, for each day of the multi-meteotsunami event, the input parameters of the stochastic surrogate model are then manually extracted from the AdriSC WRF 1.5-km modelled atmospheric disturbances along the transects selected in Section 5. The probabilities of the maximum elevation surpassing the flooding thresholds in the Vela Luka, Stari Grad and Vrboska harbours, where flooding occur during the 11-19 May 2020 period, are then determined and the meteotsunami hazards assessed for each separate event.

## 3 Description of the event and background analysis

This long-lasting meteotsunami event was reported by media, in particular by eyewitnesses in Vrboska with two YouTube videos (https://www.youtube.com/watch?v=vz9G5E9ravc; https://www.youtube.com/watch?v=-aD9q4QMANE) and by local web portals in Vela Luka and Stari Grad. In particular, Dalmacija danas (https://www.dalmacijadanas.hr/) wrote on the 14[th] of May: "Changes in air pressure have a pronounced effect on the sea-level in the Adriatic, which is most noticeable on the Dalmatian islands in the last two days. There is a constant change in sea-level throughout the day, and today it was most pronounced in the afternoon in Vela Luka. […] the sea-level fluctuated in the range of about 70 centimetres. The sea rose and flooded the waterfront, then receded abruptly, leaving the boats dry. The phenomenon was also recorded on Hvar, for example in Stari Grad, but it was less pronounced." On May 16[th] another local web portal, Morski.hr, published an article titled "Meteotsunami in Vela Luka: The sea is pouring into shops and cafes. This has been going on for three days now!" with the testimony of a local, Ljubo Padovan: "This is something we haven't had in years and it has been going on for full three days. The sea got into some shops and cafes again. When the sea recedes, one can walk from one side of the bay to the other. The sea flooded everything again last night. The situation is not calming down even after three days, that is very unusual."

Concerning the observations (Fig. 2), on the 11th of May intense high-frequency sea-level oscillations reach up to 80 cm of height (crest-to-trough) and 16 min period at 9:40 UTC in Vela Luka as well as 53 cm of height and 18 min of period at 11:07 UTC in Stari Grad. Additionally, all microbarographs record an intensification of the air pressure oscillations, with a maximum high-frequency amplitude of 3.1 hPa and period of a 13 min documented for Vela Luka. Air pressure oscillations calm down on May 12[th] and 13[th] but, following reported flooding, increase again on May 14[th], especially in Ancona and Vieste. On this day, in Vela Luka, air pressure oscillations are about 2 times weaker than on the 11th but the height of sea-level oscillations almost reaches 80 cm with a 15 min period. However, in Stari Grad, sea-levels oscillate between -25 and 25 cm from 8:00 UTC to 16:00 UTC. Even though the sea-level oscillations in Stari Grad harbour are two times smaller than during the 11[th] of May, flooding still occurred probably due to the additional effects of tidal elevation and/or storm surge. Lower intensity oscillations of both air pressure and sea-level follow the next days until around midnight on the 16th of May, when another meteotsunami event takes place. For this event, air pressure oscillations are unusually low, even in Ancona which records the

strongest ones. However, in Vela Luka the height of the sea-level oscillations goes up to 80 cm with a 13 min period. Despite the reports of flooding in Stari Grad, identically to the 14th of May, the sea-levels only oscillate between -25 and 25 cm.

The pressure oscillations do not completely vanish in the following days and on the 19th of May strong air pressure disturbances
with heights of above 2.5 hPa occur in Svetac, Vieste, Vis, Vrboska and Vela Luka. However, no flooding is recorded in Vela Luka nor Stari Grad where the recorded sea-level oscillations do not surpass 30 cm and 20 cm respectively.

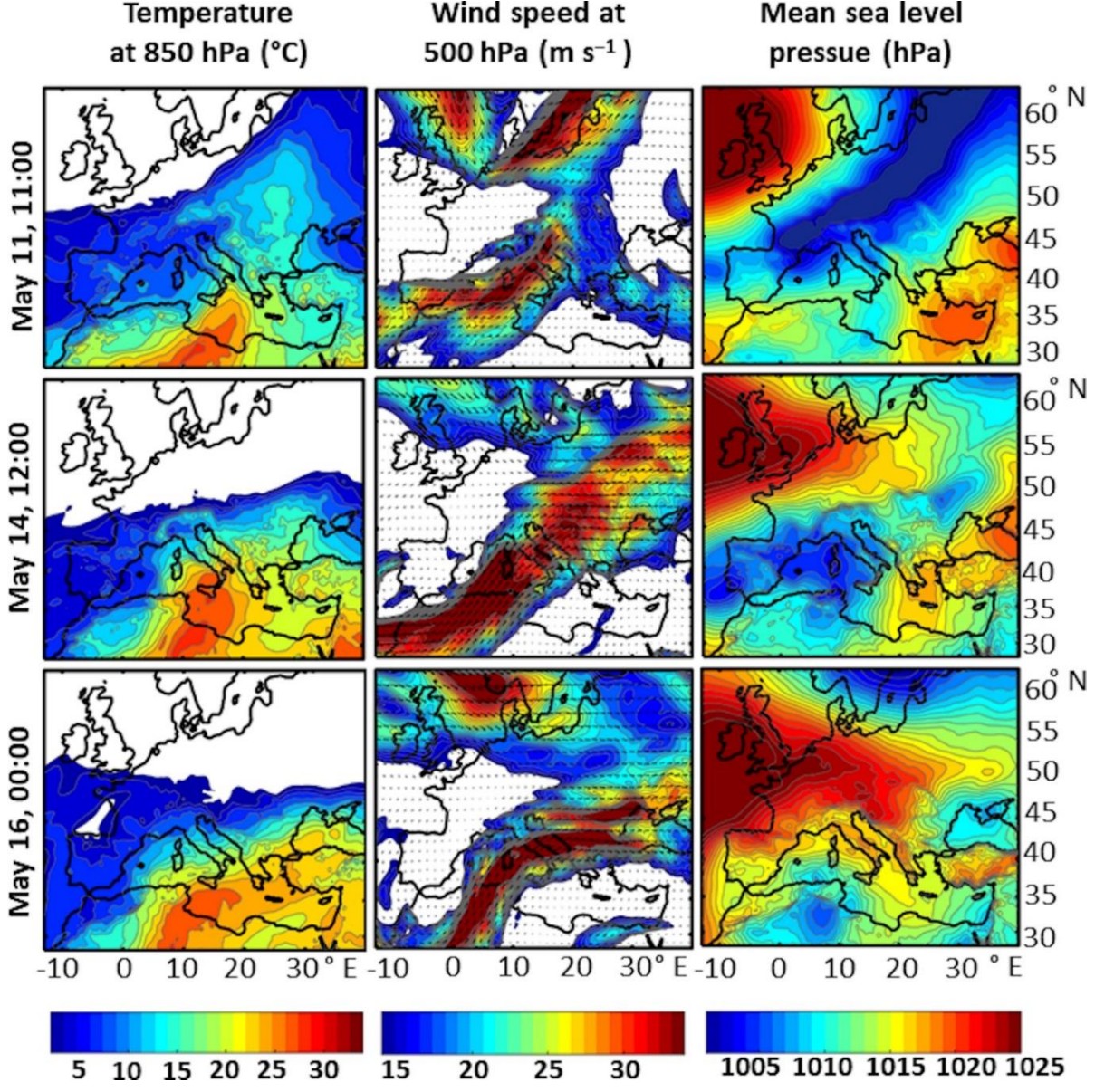

**Figure 3: Synoptic settings over Europe – temperature at 850 hPa (left panels), winds at 500 hPa (middle panels) and mean sea-level pressure (right panels) – extracted from ERA5 reanalysis at times closest to the flooding in Vela Luka, Stari Grad and Vrboska**
**harbours.**

The synoptic conditions over Europe derived from ERA5 reanalysis (Hersbach et al., 2020) – temperature at 850 hPa, winds at 500 hPa and mean sea-level air pressure (Fig. 3) – are extracted at times close to the flooding of Vela Luka, Stari Grad and Vrboska harbours: in the morning of the 11[th] of May, around midday on May 14[th] and around midnight on May 16[th]. For all the flooding events, the conditions show the advection of warm air from the Sahara towards the Adriatic at 850 hPa, associated with strong south-westerly winds at 500 hPa with speeds over 30 m s[-1]. Additionally, the mean sea-level air pressure over the Adriatic indicate either a trough stretching from the north Europe, as on the 11th, or a cyclone, deeper on the 14[th] of May and quite weak on the 16[th] of May. These synoptic conditions are known to occur during the Mediterranean meteotsunamis (Jansá et al., 2007; Vilibić et al., 2008; Šepić et al., 2016), where atmospheric disturbances (particularly atmospheric gravity waves) can be generated along the strong frontal gradients of the jet-streams as seen in Fig. 3 and propagate over long distances in the form of so-called ducted waves (Lindzen and Tung, 1976; Monserrat and Thorpe, 1996).

## 4 Measured and modelled meteotsunamigenic disturbances

The capacity of the AdriSC WRF 1.5-km and ADCIRC models to reproduce the meteotsunami events during the 11-19 May 2020 period is first assessed qualitatively by comparing the observed (in red) and modelled (in blue) time series presented in Fig. 2. It shows that the events on the 11[th] and 16[th] of May are completely missed by both the WRF 1.5-km and ADCIRC models. However, the meteotsunami event of the 14[th] of May is partially captured by the AdriSC model. Air pressure oscillations are indeed simulated in Vieste, Vis, Stari Grad, Vela Luka and Vrboska, along with weaker than measured sea-level oscillations in Stari Grad and Vela Luka. AdriSC model results for the 12[th] and 13[th] of May are generally in accordance with the measurements, with no strong oscillations of pressure and sea-level, but with slightly underestimated pressure and sea-level oscillations in Ancona, Vieste and Stari Grad and overestimated pressure oscillations in Ortona. The model results for 17[th] of May are also generally in accordance with the measurements, with underestimated pressure and sea-level oscillations in Ortona, Stari Grad and Vela Luka and overestimated pressure oscillations in Svetac. Even though the deterministic AdriSC model fails to forecast two of the three observed meteotsunami events, the event mode of the CMeEWS is triggered for all the days of the 11-19 May 2020 period except for the 12[th] and 13[th] for which no false alarms would have been triggered (Fig. S1 provided as supplementary material).

Measured and modelled composites of air pressure frequency-time spectrograms in the Eastern, Middle and Western Adriatic Sea (Fig. 4 to 6) are thus used to quantitatively compare the energy content of the meteotsunamigenic disturbances as observed by the microbarographs and forecasted with the WRF 1.5-km model at grid points W1-W7, M1-M2 and E1-E6, in addition to grid points next to microbarograph stations as described in Section 2. Overall, no pronounced energy peaks are found in the spectrograms, which is typical for spectra of air pressure characterized by a number of oscillatory movements with no dominant period (Monserrat and Thorpe, 1992; Zemunik et al., 2020). Additionally, in operational mode, the CMeEWS would have provided warnings for a full day (next 30-h period including night hours pass midnight) and not for a precise time. It thus may

be noticed that, despite this analysis being temporal, discussions about the differences between modelled and measured timing of the meteotsunami events are not relevant for the model verification.

For the 11th of May, the highest energies from the observed composite are located at Ortona with frequencies below 1.5 10-3 Hz (11 min period) and around 1.8 10-3 Hz (9.25 min period) for the western Adriatic region, at Svetac with frequencies below 1.0 10-3 Hz (16.5 min period) for the middle Adriatic region, and at Vela Luka with frequencies below 1.1 10-3 Hz (15 min period) as well as with the 1.4 10-3 Hz (12 min) and 1.9 10-3 Hz (8.8 min) frequencies for the eastern Adriatic region. For this event, the WRF 1.5-km model produces at E1 located far northwest from Vela Luka substantially lower energies at the same frequencies than the observed composite, but with high energies at frequencies up to 1.1 10-3 Hz (15 min period) in the western Adriatic, up to 0.8 10-3 Hz (20.8 min period) in the middle Adriatic and at 1.0 10-3 Hz (16.5 min period). This implies that the modelled atmospheric disturbances are less energetic and located further north compare to the observed ones.

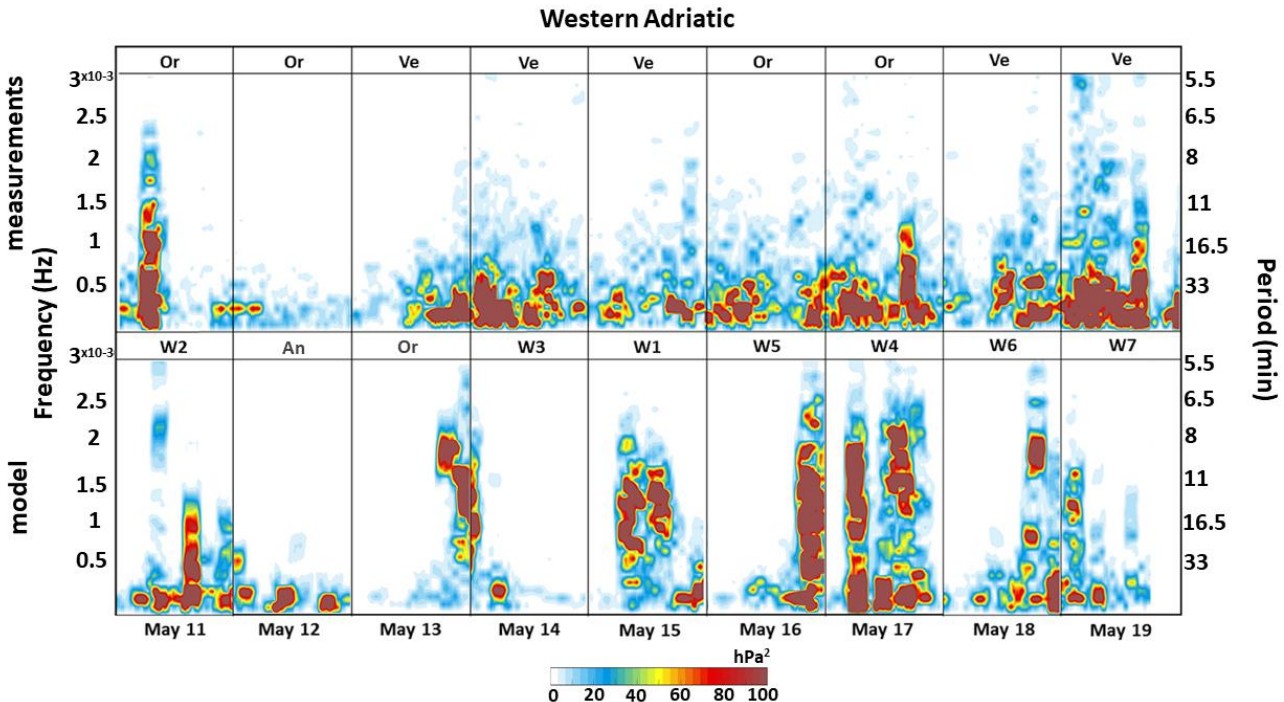

**Figure 4: Modelled and measured composites of high-pass filtered air pressure frequency-time spectrograms for the western Adriatic region. Maximum daily energies measured by the microbarographs (observed composite) and modelled at one WRF 1.5-km model grid point (modelled composite) are collocated.**

During the calm period between the 12th and 13th of May, the energy of the observed spectrograms is much lower than during the meteotsunami events. The model produces extremely low energies for both days in all regions, with high energies at frequencies up to 2.0 10-3 Hz only in Ortona, in the western Adriatic region, on the evening of 13th of May.

However, on the 14[th] of May, the highest energy values from the observed composite are found at Vieste with frequencies

below 0.7 10-3 Hz (24 min period) for the western Adriatic region, at Vis with frequencies below 0.55 10-3 Hz (30 min period) for the middle Adriatic region, and at Stari Grad with frequencies below 0.5 10-3 Hz (33 min period) for the eastern Adriatic region. The highest energies simulated by the model are located at W3 with frequencies up to 1.8 10-3 Hz (9.25 min period) for the western Adriatic region, at Vis with frequencies up to 1.1 10-3 Hz (15 min period) for the middle Adriatic region, and at E6 located south from Stari Grad with frequencies up to 1.5 10-3 Hz (11 min period) for the eastern Adriatic region. It is

unlikely that the modelled atmospheric disturbance can travel from W3 to E6 by crossing diagonally the middle Adriatic region. The results are thus probably coming from more-than-one atmospheric disturbances that occurred during the 14[th] of May and changed their energies when crossing the Adriatic.

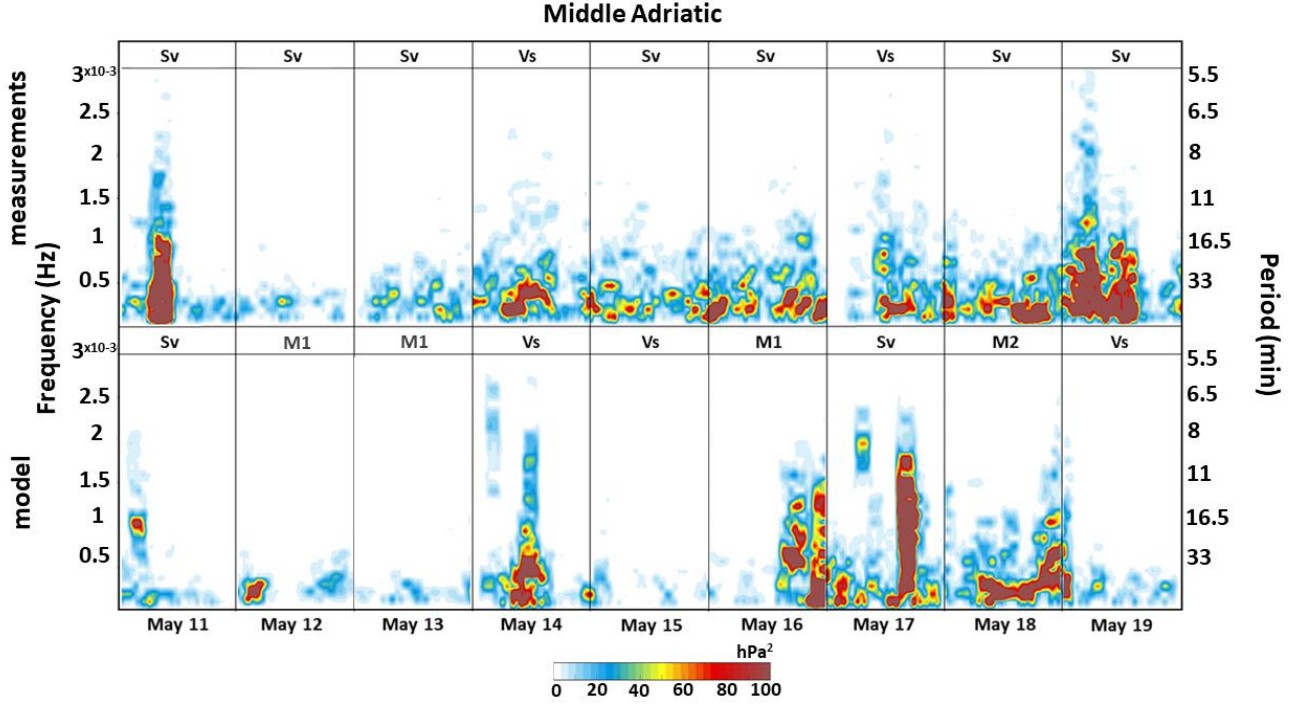

**Figure 5: As in Fig. 4, but for the middle Adriatic region.**

For the 15[th] and 16[th] of May, the energies of the observed composite are higher than during the calm background period (between the 12[th] and 13[th] of May), but lower than during the meteotsunami events. For the 15[th] of May, the model produces high energies at frequencies up to 2.0 10-3 Hz (8 min period) at W1 in the western Adriatic region. However, this disturbance does not propagate to the east as the spectrograms in the middle and eastern Adriatic regions have extremely low energy. Energy in the model for the 16[th] of May is negligible in the western Adriatic region, but for the middle and eastern Adriatic

regions high energies at frequencies below 1.5 10-3 Hz are found at M1 and E2, respectively. In other words, even though the

meteotsunami event of the 16th of May is missed by the AdriSC model (Fig. 2), the WRF 1.5-km model simulates a strong meteotsunamigenic disturbance shifted north-westward compared to the observations.

On the 17th of May the highest energies from the observed composite are located at Ortona for the western Adriatic region, at Vis for the middle Adriatic region and at Stari Grad for the eastern Adriatic region. High energies are also found in the modelled
composite at up to 2.2 10-3 Hz (7.5 min period) at W4 for the western Adriatic region, up to 1.8 10-3 Hz (9 min) at Svetac for the middle Adriatic region, and up to 1.0 10-3 Hz (16.5 min period) at E5 for the eastern Adriatic region. The spatial layout of the highest energy points again illustrates the limitations of the applied methodology when multiple disturbances are simulated. Nevertheless, obtained results imply that the different disturbances in the model are more energetic than the one observed by the microbarographs.

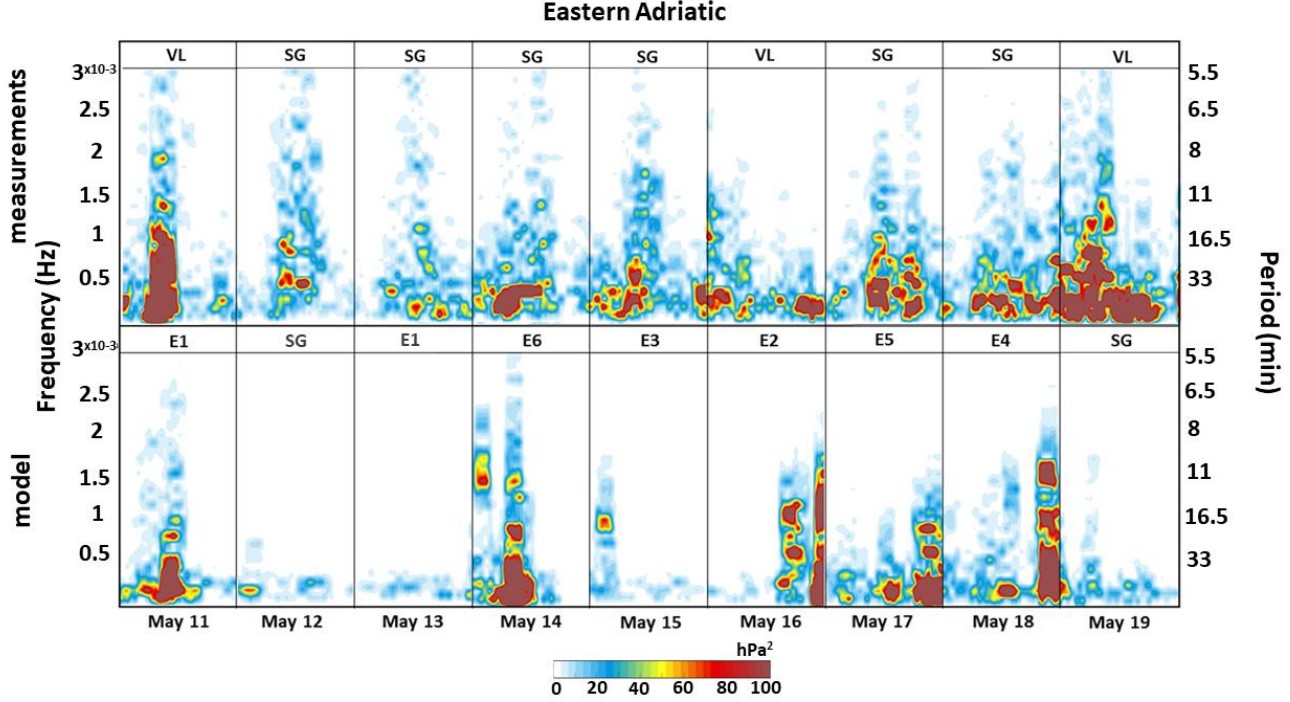

**Figure 6: As in Fig. 4, but for the eastern Adriatic region.**

More energy is found in the observed spectrograms for the 18th of May than during the calm background period, but less than during the meteotsunami events. The model, however, produces high energies at high-frequencies of about 2.0 10-3 Hz (8 min period) at W6 for the western Adriatic region and 1.75 10-3 Hz (9.5 min period) at E4 for the eastern Adriatic region.
Atmospheric disturbance energy at M2 is the highest for middle Adriatic region than for the western and the eastern Adriatic regions, particularly for lower frequencies.

Conditions are again more energetic on the 19$^{th}$ of May, with highest energies in the observed composite at frequencies up to 1.5 10-3 Hz (11 min period) at Vieste and Vela Luka, and up to 1.25 10-3 Hz (13 min period) at Svetac. The model fails to reproduce these disturbances and only simulates high energies at W7, in the southernmost point of the studied area.

In brief, periods between 10 min and 20 min – typical of meteotsunamigenic disturbances – are found to often occur in the analysis of the frequency-time spectrogram composites (Figs. 4 to 6). Additionally, systematic biases exist in the forecasted atmospheric disturbances, as often simulated further northwest than the observed ones. Finally, this analysis has demonstrated that the Adriatic high-frequency sea-level oscillations of 11-19 May 2020 are induced by atmospheric forcing of diverse spatial and temporal characteristics.

## 350    5 Meteotsunami energy banners

Given the lack of reliability of the deterministic AdriSC model to properly forecast spatial and temporal characteristics of the multi-meteotsunami event of the 11-19 May 2020 period, the warnings released by the CMeEWS would have fully rely on the results of the stochastic surrogate model forced with input parameters extracted from the WRF 1.5-km simulations. The values of the six stochastic parameters – which serve as input of the stochastic surrogate model – are derived from the modelled

meteotsunamigenic disturbances. The meteotsunami energy banners including their impact to the ocean are thus documented along the selected transects where these parameters are extracted. As described in Section 2, the operational sampling criterion (hereafter referred as air pressure variance), the new transect sampling criterion, the atmospheric and ocean spectrograms as well as the Proudman resonance along the most energetic transects are displayed in Figs. 7 to 11 and in supplementary material (Figs. S2 to S15) for each day of the multi-meteotsunami event.

For the 11$^{th}$ of May, the modelled air pressure variances (Figs. 7, S2 and S3, left top panel) and the associated new transect sampling criterion (Figs. 7, S2 and S3, right top panel) are indicating maximum meteotsunami energy banners located far too northwest from Vela Luka, Vrboska and Stari Grad harbours, where the meteotsunami event is observed. Nevertheless, atmosphere over the two selected transects is highly energetic and the pronounced disturbances are travelling with a speed between 12.5 m s−1 and 33.32 m s−1 over relatively shallow areas. Despite the Proudman resonance being possible over a

large section of the transects, the energy transferred to the ocean is not substantial anywhere but near the coast.

For the 14$^{th}$ of May, several modelled atmospheric disturbances are located in the middle Adriatic region (Figs. 8 and S4 to S6, left top panel). The location of the highest air pressure variances and the associated new transect sampling criterion (top panels, Fig. 8), as well as the speed of the tracked most energetic disturbance of 27.9 m s$^{-1}$, makes this disturbance a good candidate for causing the meteotsunamis that flooded Vela Luka, Vrboska and Stari Grad harbours on this day. Nevertheless,

the transect is in a deep water with changing bathymetry, and therefore the Proudman resonance is only likely to happen over a small part of the transect, while other effects, including edge waves, strong topographical enhancement and refractions on the islands placed on the pathway of atmospheric disturbances may be important for generation of meteotsunami waves in the middle Adriatic (Šepić et al., 2016). Higher energies in the atmosphere, but not in the ocean, can be found on spectrograms of

transects in Figs. S4 and S6. These disturbances are located too south or too north of the domain to cause meteotsunamis in the harbours of interest. Also, the speeds of the tracked disturbances in Figs. S4 and S5 are not within the range of speeds of meteotsunamigenic disturbances.

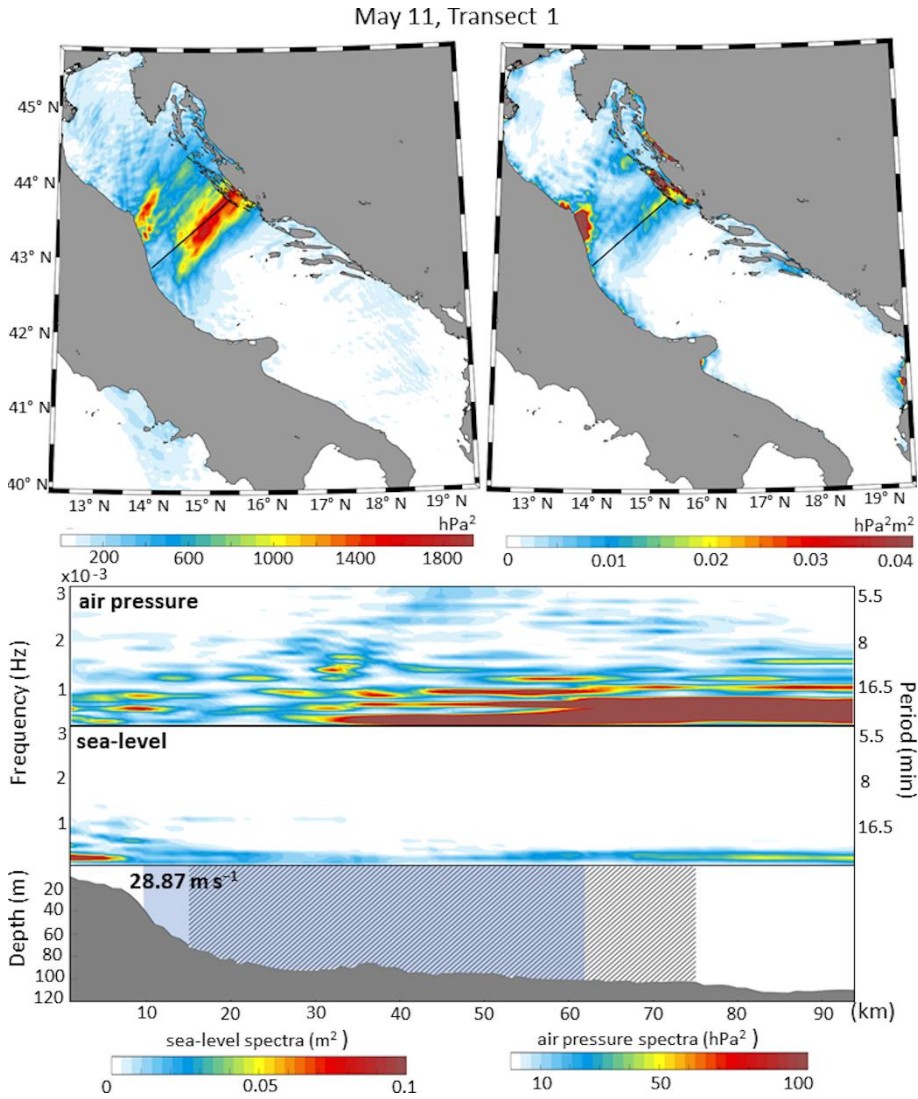

**Figure 7: Meteotsunamigenic disturbance of the 11th of May 2020 along Transect 1. Air pressure spatial variance (top left panel) and new transect sampling criterion (top right panel) have a mark of the selected transect containing meteotsunami energy banner (solid black line). Spectrograms of high-pass filtered mean sea-level air pressure (air pressure) and sea-level along the selected transect (middle panels) are conjoined by sections of the associated depth profile (bottom panel) where the Proudman resonance is likely to occur (shaded with diagonal stripes) and where the speed of the disturbance is calculated (in blue).**

Two atmospheric disturbances are tracked for the 15th of May and presented in Figs. 9 and S7. The maximum in air pressure variance (Fig. 9, left top panel) and the associated maximum in new transect sampling criterion (Fig. 9, right top panel) are

385 located too northwest to cause the Vela Luka and Stari Grad flooding in the night of the 15$^{th}$ to 16$^{th}$ of May. Also, despite the high energies in the atmosphere, no transfer to the sea can be seen along the transect, being restricted just near the coast. This is probably due to the low speed of the disturbance (i.e. 10.5 m s$^{-1}$) and the depth (i.e. over 100 m) along the transect. Spectrograms in Fig.S7 display high energies and strong ocean response at the beginning of the transect, but negligibly small energy values on the rest of the transect, which is a good example of a dissipating disturbance. Low speed of the atmospheric

disturbance of only 11.6 m s$^{-1}$ and the lack of flat seabed could explain such a behaviour.

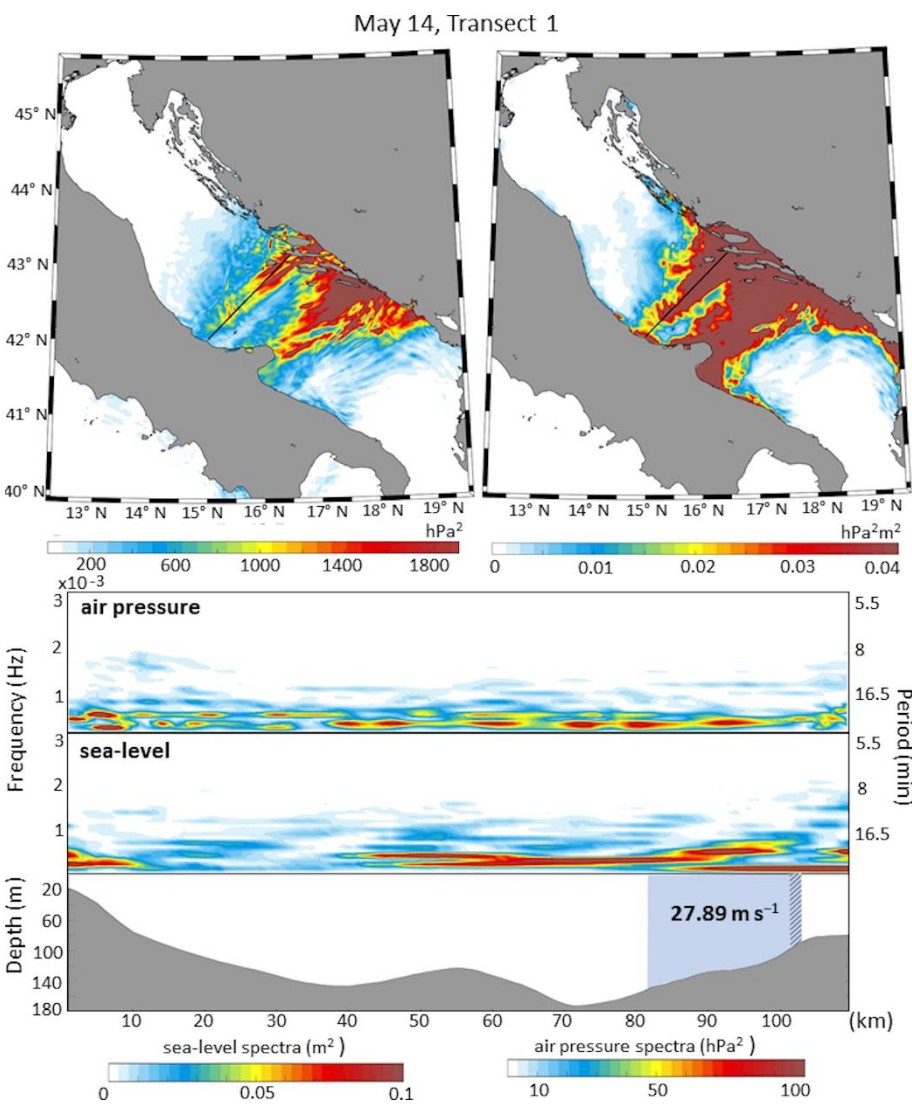

**Figure 8: As in Fig. 7, but for the meteotsunamigenic disturbance of the 14$^{th}$ of May 2020 along Transect 1.**

Three disturbances are analysed for 16$^{th}$ of May and presented in Figures 10, S8 and S9. Two northwestwardly shifted atmospheric disturbances (Figs. 10 and S8) are extremely energetic and the transfer of energy to the sea is strong at the

beginnings of the transects. Speeds of the disturbances, of 7.1 m s⁻¹ and 11.3 m s⁻¹, are low compared to the normal speeds for meteotsunamigenic disturbances. Southern disturbance (Fig. S9) has greater speed, of 25 m s⁻¹, but neither the atmosphere is highly energetic nor the transfer of energy to the sea is strong anywhere but near the coast. This is displayed in both top panels, and in spectrograms of Fig. S9. It should be noticed that the air-sea interaction is the strongest over the area where Proudman resonance is likely to happen.

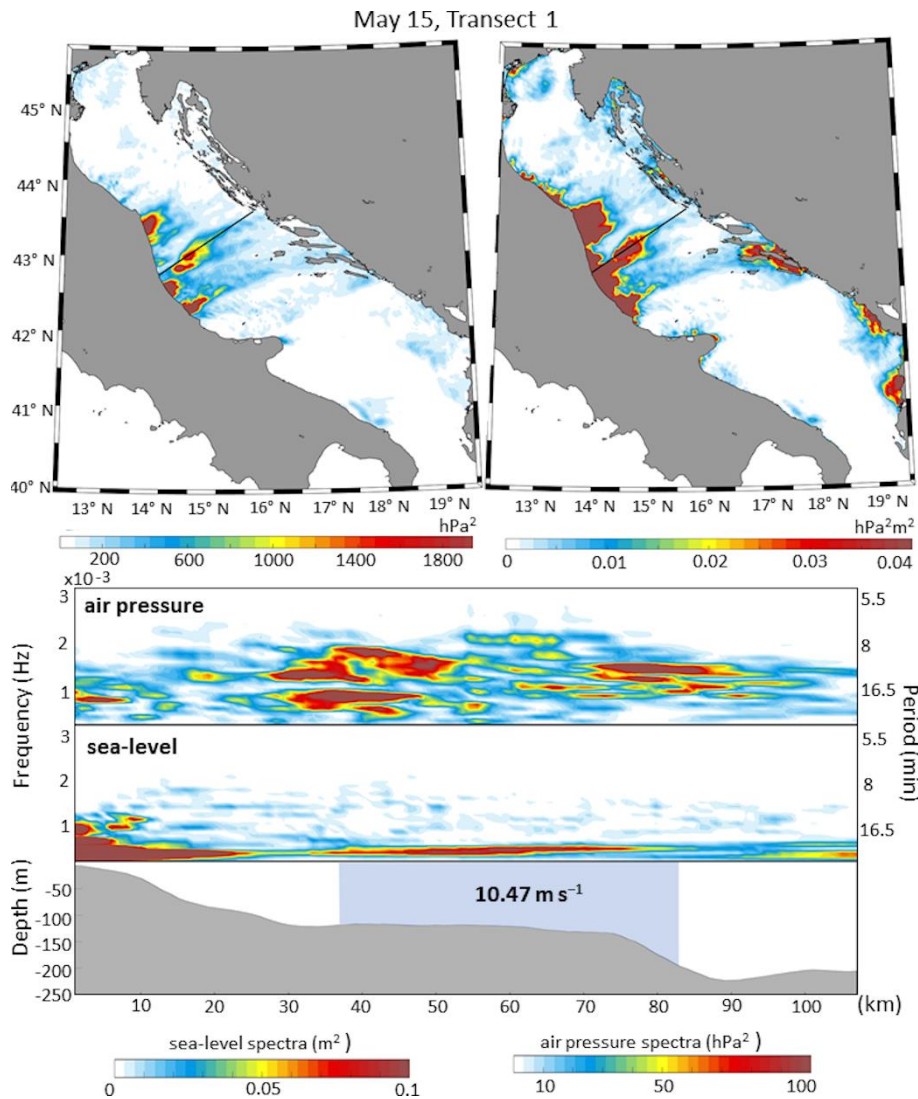

**Figure 9: As in Fig. 7, but for the meteotsunamigenic disturbance of the 15th of May 2020 along Transect 1.**

For the 17th of May, two of three modelled atmospheric disturbances (Figs. S10 and S11) are located where they could cause meteotsunamis along the eastern Adriatic coastline. However, the speeds of these disturbances, ranging from 10.3 m s⁻¹ to 12.1 m s⁻¹, are too low and the atmosphere and the sea are not as energetic as they are over the transect analysed in Fig. 11. The

405 atmosphere is extremely energetic over the selected transect and, since energy is well transferred to the ocean, high energies occurred for high frequencies in the ocean too. Spectrograms in Fig. 11 show that ocean's response to atmospheric disturbance is pronounced over the whole transect, but is the strongest over the section which satisfied the Proudman resonance conditions. The disturbance travelled with 27.8 m s$^{-1}$, but as seen in top panels, it is again located in the northern part of middle Adriatic.

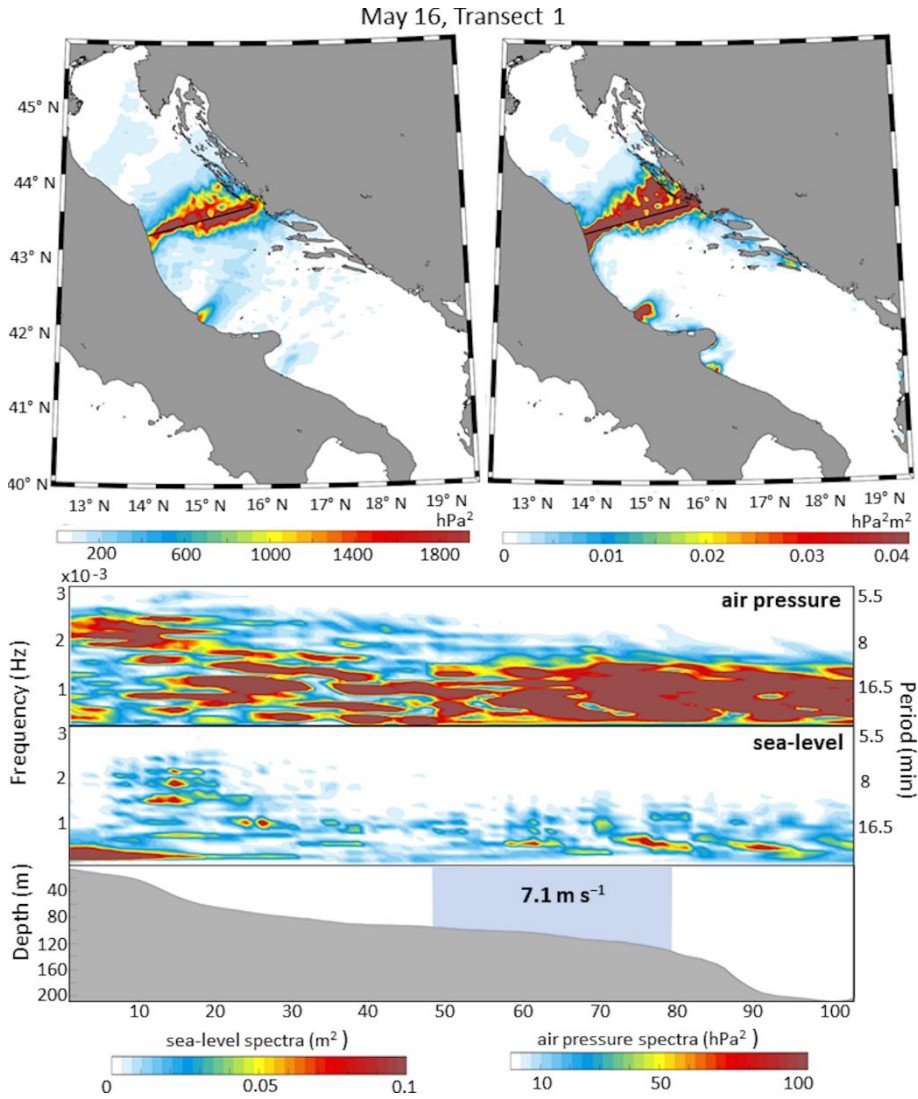

**Figure 10: As in Fig. 7, but for the meteotsunamigenic disturbance of the 16$^{th}$ of May 2020 along Transect 1.**

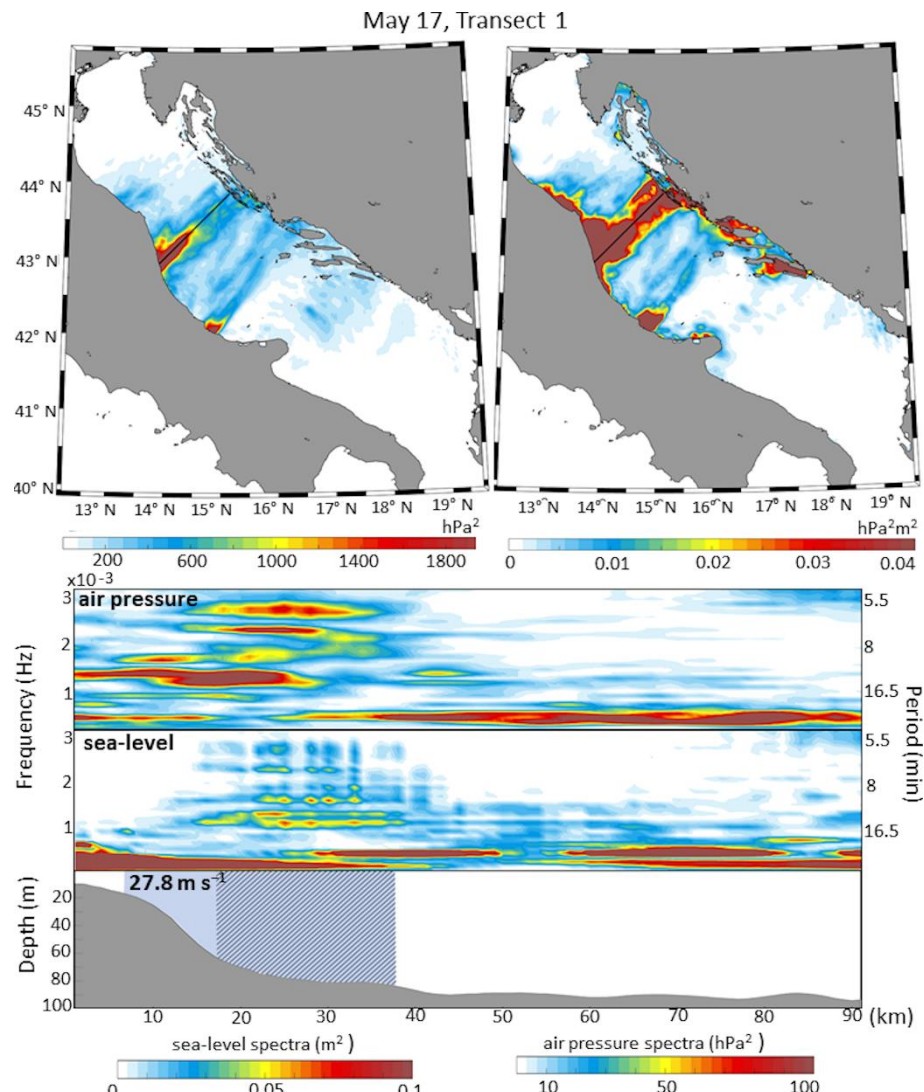

**Figure 11: As in Fig. 7, but for the meteotsunamigenic disturbance of the 17th of May 2020 along Transect 1.**

For the 18th of May, the modelled atmospheric disturbances (Figs. S12 to S14) are crossing the middle Adriatic from southwest to northeast, over the common path of meteotsunamigenic disturbances. Speeds of the tracked disturbances vary from 20.2 m s⁻¹ to 30.3 m s⁻¹. Even though atmosphere is energetic for the transects presented in Figs. S12 and S14, the energy of the sea is not significantly higher for them than for the transect in Fig. S13, with low energy in the atmosphere. Therefore, despite the appropriate speeds and locations of the meteotsunamigenic disturbances, the energy is not well transferred from the atmosphere to the sea and no meteotsunami event is modelled.

For the 19$^{th}$ of May there is only one modelled disturbance, travelling at 19.4 m s$^{-1}$ far south of the analysed region (Fig. S15).
Energy content of both atmosphere and the sea is low for the selected transect, but some air-sea interactions are taking place in the eastern end of the transect (right top panel, Fig. S15).

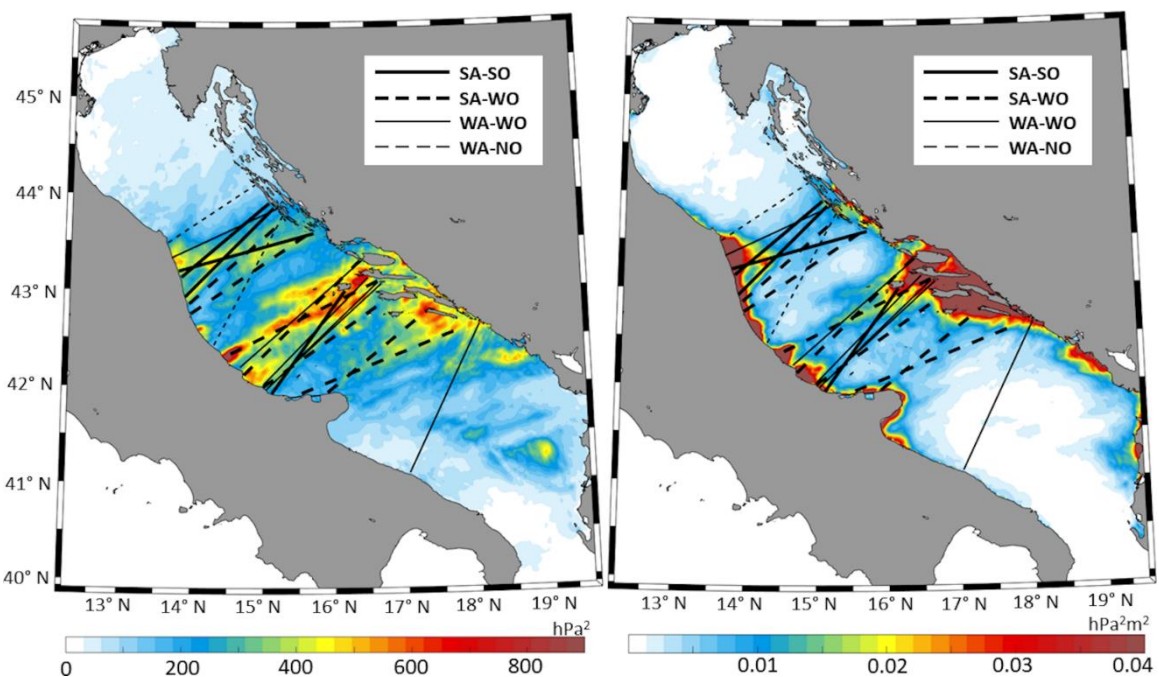

**Figure 12: Averaged air pressure variances (left panel) and averaged new transect sampling criterion (right panel) estimated for the ensemble of transects extracted between the 11$^{th}$ and 19$^{th}$ of May 2020. The modelled disturbances (black lines) are categorized depending on the strength of the atmospheric signal (SA for strong and WA for weak) and the ocean response (SO for strong, WO for weak, NO for none).**

Averaged air pressure variance and averaged new transect sampling criterion derived from all the extracted transects between the 11$^{th}$ and 19$^{th}$ of May 2020 are presented in Fig.12, together with the selected transects. The transects are classified in four different categories emphasizing the strength of the atmospheric disturbances as well as the energy transfer from the atmosphere to the ocean. The most intense atmospheric activity and air-sea interactions are located across the middle Adriatic region. Additionally, despite relatively low averaged air pressure variances, the averaged values of the new transect sampling criterion are the highest along the Dalmatian island and the middle Adriatic Italian coastlines. These results thus confirm that the intensity of the atmospheric disturbances is less important than the resonance – i.e. appropriate speed, period and depth along the transects (Denamiel et al., 2020) – and, of course, the bathymetry.

In brief, the presented results of the travelling air-sea meteotsunami energy banners show that the ocean model response to the atmospheric forcing highly depends on both the location and the frequency of the meteotsunamigenic disturbances, which are in our study often modelled too northwest of the most affected locations. Finally, the introduced new transect sampling criterion does not seem to overall facilitate the decision-making process in terms of the transect selection, since all the transects selected

by this criterion would have also been selected following highest values of air pressure variances only. Even though for some events (e.g. Figs. 9, 10, 11) the new criterion highlights the strength of the air-sea interactions, these interactions are located along the same transects as captured by the highest values of the air pressure variance. As efficiency is important in early warning system, it can thus be concluded that the use of the ocean model results to better select the transect with maximum meteotsunami generation is not necessary in operational mode, since it would be more time consuming with no significant value added to the process of the transects selection.

## 6 Stochastic hazard assessment

The analysis presented in previous sections has proven that the operational deterministic AdriSC model is not capable to properly reproduce the meteotsunami events of the 11-19 May 2020 period. However, parameters like location, amplitude, direction, speed, period, and width can be extracted from the atmospheric disturbances produced by the WRF 1.5-km model and used as inputs of the stochastic surrogate model. For the 11-19 May 2020 period (with the exception of the 12th and 13th), the stochastic surrogate model is thus run for Vela Luka, Stari Grad and Vrboska, with input variables from the atmospheric disturbances selected for each day along the transects presented in the previous section. The probabilities of the maximum elevation surpassing the flooding threshold are presented in Table 2.

**Table 2. Meteotsunami hazard assessment derived with the stochastic surrogate model and provided as the probability of maximum sea elevation crossing the flooding thresholds in Vela Luka, Stari Grad and Vrboska during the 11-19 May 2020 period.**

| Location | Probability of crossing the flooding threshold (%) during the 11-19 May 2020 multi-meteotsunami event | | | | | | |
|---|---|---|---|---|---|---|---|
| | 11th | 14th | 15th | 16th | 17th | 18th | 19th |
| Vela Luka | *16* | *10* | *19* | *14* | *10* | 4 | *34* |
| Stari Grad | *13* | *2* | *6* | *6* | *4* | *1* | *9* |
| Vrboska | *16* | *18* | 19 | 23 | 22 | 3 | 37 |

Note: When the probabilities are above or equal to 10 % (highlighted in bold), the meteotsunami warning is triggered. In addition, probabilities at locations at which flooding has been reported by eyewitnesses during the events are highlighted in italics.

For the 11-19 May 2020 period when flooding and strong sea oscillations were reported for Vela Luka, Stari Grad and Vrboska (in italics, Table 2), the meteotsunami warning would have been triggered in Vela Luka and Vrboska for all the events, but only for the 11th of May in Stari Grad. The results found in Stari Grad are, however, in good agreement with the moderate oscillations (amplitude of 25-30 cm) of the high-pass filtered sea-levels extracted between the 14th and the 16th of May at the tide gauge location. Additionally, the meteotsunami warning would have been wrongly triggered the 17th and the 19th in Vela Luka and the 15th, 16th, 17th and 19th in Vrboska when no flooding was reported. It is worth noticing that, for the 17th and 19th of May 2020, the forecasted meteotsunamigenic conditions capable to trigger the event mode of the CMeEWS are, in fact, in good agreement with the strong air pressure oscillations observed along the western Adriatic coast (Fig. 2). Additionally, as

already shown in Denamiel et al. (2019b), false alarms are easily triggered in Vrboska. This may be linked to either the poor representation of the Vrboska geomorphology within the ADCIRC model used to create the surrogate model or the choice of the flooding threshold and therefore should be further investigated.

## 7 Summary and conclusions

In the Adriatic Sea, recurrent meteotsunami events are known to strongly impact the way of life of the coastal communities, particularly in the Dalmatian islands where they can generate serious flooding. In this study, the capacity of the Croatian meteotsunami early warning system (CMeEWS), which provides meteotsunami hazard assessments depending on the deterministically forecasted and measured air pressure disturbances and the stochastically deduced maximum elevation distributions derived with the surrogate model, is examined. As it is no longer operational, the capacity of the CMeEWS is
evaluated retroactively for the multi-event of the 11-19 May 2020. This event is of particular interest because meteotsunamigenic synoptic patterns over the Adriatic were present during a prolonged period of about 5 to 10 days, not previously observed for any meteotsunami. During this period, intense high-frequency air pressure and sea-level oscillations were observed and recorded in the middle Adriatic with maximum sea-levels reached the 11th, 14th and 16th of May in Vela Luka, Stari Grad and Vrboska.

One of the main originalities of this study is that the performances of the CMeEWS operational models – i.e. the WRF 1.5-km atmospheric model and the ADCIRC ocean model from the AdriSC modelling suite – are assessed via energy banners. Analysis of composites of frequency-time spectrograms has shown that the deterministic models are generally not capable to reproduce the meteotsunami events in affected bays but can produce strong meteotsunamigenic disturbances often shifted north-westward from them. It was demonstrated that, even though the strongest atmospheric activity was modelled in the middle Adriatic along
common air pressure disturbance pathways, the meteotsunami events were always missed by the ADCIRC ocean model at Vela Luka and Stari Grad during the 11-19 May 2020 period due to a shift in location of the modelled atmospheric disturbances. This most probably indicates that the frequency of the air pressure disturbances is not properly reproduced by the WRF 1.5-km model, posing a question of appropriateness of the state-of-the-art atmospheric models in terms of their resolution and setup (Horvath and Vilibić, 2014). Finally, this study also highlighted that using the ocean model results in combination with
the atmospheric model results with the so-called new transect sampling criterion is not helping to improve the selection of atmospheric conditions needed to feed the stochastic meteotsunami surrogate model. However, due to the systematic error link to the shift of the disturbances towards the north, it may be envisioned in the future to apply a correction concerning the start point location before using the surrogate model.

Given these results, the following question can be raised: should the ADCIRC ocean model be run in operational mode within
the CMeEWS or should the meteotsunami hazard assessments be derived solely with the stochastic surrogate model? In the presented case, as deterministic ocean model fails for all events due to a shift in location of the modelled atmospheric disturbances, the question is easily answered. And, in general, due to the uncertainties associated with operational modelling

of meteotsunamigenic disturbances, the stochastic approach has proven to be an optimal option. Nevertheless, the ADCIRC ocean model can still be used for other hazards such as extreme storm surges associated with wind-waves and not only for meteotsunami events.

Concerning the evaluation of the stochastic model fed by the extracted meteotsunamigenic air pressure conditions along the selected transects, in most of the cases and despite some false alarms, the coastal communities of Vela Luka, Stari Grad and Vrboska would have been warned of potential meteotsunami events if the CMeEWS had been operational. Even though warning effectiveness highly depends on the resident trust which can be easily eroded from false alarms and/or missed events, the uncertainty faced by the Croatian coastal communities during the 11-19 May 2020 period and reported by several local newspapers, is probably far worst. The meteotsunami surrogate model, even if not perfect as not including the storm surges in Stari Grad for example, has thus been proven to be extremely useful and reliable during this multi-meteotsunami event.

The complexity of forecasting the precise location, intensity, speed, etc. of the atmospheric disturbances triggering the most extreme sea-level events around the world is one of the biggest issues faced by the meteotsunami community. In consequence, different approaches have been recently implemented within the two meteotsunami early warning systems existing in the Mediterranean Sea (Denamiel et al. 2019, Mourre et al., 2020, Romero et al., 2020). To conclude, as operational models often fail to properly forecast extreme events, the continuous development of stochastic approaches – such as the meteotsunami surrogate model within the CMeEWS – described in Denamiel et al. (2021) should be an avenue explored by the extreme sea-level community in order to improve early warning systems.

**Code availability**

Codes of COAWST, WRF and ADCIRC models, can be obtained on the following links: https://www.usgs.gov/software/coupled-ocean-atmosphere-wave-sediment-transport-coawst-modeling-system, https://www2.mmm.ucar.edu/wrf/users/downloads.html, http://adcirc.org/.

**Data availability**

The model results and the measurements used to produce this article can be obtained under the Open Science Framework (OSF) FAIR data repository https://osf.io/24m8e/ (doi: 10.17605/OSF.IO/24M8E).

**Author contributions**

IV and CD defined concept and design of the study. Material preparation was done by CD and IT. Set-up of the model and simulations were performed by CD and IT. Production of the figures was done by CD and IT. Analysis of the results was

performed by IV, CD and IT. The first draft of the manuscript was written by IT. All authors were engaged in commenting, revising and polishing of the manuscript. All authors read and approved the final manuscript.

**Competing interests**

The authors declare that they have no conflict of interest.

**Acknowledgements**

Special thanks are given for the support of the European Centre for Middle-range Weather Forecast (ECMWF) staff, in particular Xavier Abellan and Carsten Maass, as well as for ECMWF's computing and archive facilities used in this research. This work has been supported by projects ADIOS (Croatian Science Foundation Grant IP-2016-06-1955), BivACME (Croatian Science Foundation Grant IP-2019-04-8542), CHANGE WE CARE (Interreg Croatia-Italy program) and three ECMWF Special Projects (The Adriatic decadal and inter-annual oscillations: modelling component, Numerical modelling of the

Adriatic-Ionian decadal and inter-annual oscillations: from realistic simulations to process-oriented experiments, and, Using stochastic surrogate methods for advancing towards reliable meteotsunami early warning).

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
