# Peer review of "Performance of the Adriatic early warning system during the multimeteotsunami event of 11-19 May 2020: an assessment using energy banners"

_Natural Hazards and Earth System Sciences, 2020_

## Author Response (AR1)

**Response to the reviewers' comments on the manuscript "Performance of the Adriatic early warning system during the multi-meteotsunami event of 11-19 May 2020: an assessment using energy banners" by Tojčić et al., submitted to NHESS (nhess-2020-409)**

The authors would like to thank both anonymous reviewers and the editor for their detailed comments which helped to largely improve the new version of the article.

**Editor**

*"- In the introduction, you briefly comment whether dynamics, intensity and consequences of meteo-tsunamis are similar or significantly different in the geographical examples that you mention."*

**Response:** The following sentence has been added in the introduction: "For all these locations despite varying intensities, meteotsunami events have the potential to generate structural damages and sometimes even human casualties."

*"- In the conclusion, whether the problems that you encounter with the predictions are unique for the Croatian coast. Do you have reason to expect that the stochastic model will outperform the deterministic model also in other locations? Have your results significant implications for other mentioned locations?"*

**Response:** The following sentences have been added to the last paragraph of the conclusions: "The complexity of forecasting the precise location, intensity, speed, etc. of the atmospheric disturbances triggering the most extreme sea-level events around the world is one of the biggest issues faced by the meteotsunami community. In consequence, different approaches have been recently implemented within the two meteotsunami early warning systems existing in the Mediterranean Sea (Denamiel et al. 2019, Mourre et al., 2020, Romero et al., 2020). "

**Anonymous Reviewer #1 (RC1)**

*"Performance of the Adriatic early warning system during the multi-meteotsunami event of 11-19 May 2020: an assessment using energy banners" is an interesting manuscript concerned with using numerical methods to forecast meteotsunami in coastal areas. The article contains a detailed description of the forecasting system performance during the multi-meteotsunami event that hit the eastern Adriatic coast in 2020. The objective of numerical experimentation is clearly stated. I must say, however, I had some difficulties in reading the manuscript and follow the author's*

*approach. As it is now, I cannot evaluate the capacity of the modelling system in forecasting meteotsunami events.*

**Response:** Thank you for your comments. A "Methods" section is introduced in the new version of the article. We hope that it will help to clarify and simplify the article and hence to make it easier for the readers to follow the study.

*An unclear point of this study is the definition of the disturbance trajectories (transects) and associated energy banners. Part of section 5 should be moved before (or at the beginning) of section 4 in order to understand the presented results. On this topic, the authors should clear explain:*

- *how the transects are selected;*
- *the number of transect per event;*
- *how to compare model results with observation at different locations (figures 3 to 5). As they are now, these figures are not useful for understanding the model performance;*
- *what's the temporal rate of change for identifying events (on which time interval);*
- *why the transect sampling criteria, which accounts for the open-ocean resonance, does not provide useful indications for selecting the transects.*

**Response:** We agree with the reviewer that the methodologies used in the article were not clearly described. We added the following "Methods" section in order to address the first four bullet points raised above:

[revised manuscript text omitted]

Regarding the fifth bullet point, we added the following clarification at the end of Section 8 (Summary and conclusions):

"Finally, the introduced transect sampling criteria does not seem to overall facilitate the decision-making process in terms of the transect selection, since all the transects selected by this criterion would have also been selected following highest values of air pressure variances only. Even though for some events (e.g. Figs. 8, 9, 10) the new criterion highlights the strength of the air-sea interactions, these interactions are located along the same transects as captured by the highest values of the air pressure variance. As efficiency is important in early warning system, it can thus be concluded that the use of the ocean model results to better select the transect with maximum meteotsunami generation is not necessary in operational mode, since it would be more time consuming with no significant value added to the process of the transects selection."

*Moreover, some aspects mainly related to the Meteotsunami Early Warning System need to be improved. The authors should provide more details about:*

- *the numerical models' implementation, e.g. model domains, grid resolution, boundary and forcing conditions;*
- *the observational systems, e.g. type of instruments, acquisition frequencies, filtering of the wind-wave effects on the tide gauge data;*
- *high-pass filtering procedure of observation and model results.*

**Response:** Accepted.

First bullet point is addressed with the following paragraph:

"The basic module uses a modified version of the Coupled Ocean-Atmosphere-Wave-Sediment-Transport (COAWST) modelling system developed by Warner et al. (2010), built around the Model Coupling Toolkit (MCT) which exchanges data fields and dynamically couples the Weather Research and Forecasting (WRF) atmospheric model, the Regional Ocean Modeling System (ROMS), and the Simulating WAves Nearshore (SWAN) model. The basic module is set-up with (1) two different nested grids of 15-km and 3-km resolution used in the WRF model and covering respectively the central Mediterranean area and the Adriatic-Ionian region and (2) two different nested grids of 3-km and 1-km resolution used for both ROMS and SWAN models and covering respectively the Adriatic-Ionian region (similarly to the WRF 3-km grid) and the Adriatic Sea only.

The dedicated meteotsunami module couples offline the Weather Research and Forecasting (WRF) model (Skamarock et al., 2005) with the 2DDI (i.e. two-dimensional depth-integrated) unstructured ADvanced CIRCulation (ADCIRC) model (Luettich et al., 1991), using a mesh of up to 10-m in resolution in the areas sensitive to meteotsunami hazard. In more details, (1) the hourly results from the WRF 3-km grid obtained with the basic module are first downscaled to a WRF 1.5-km grid covering the Adriatic Sea and (2) the hourly sea surface elevation from the ROMS 1-km grid, the 10-min spectral wave results from the SWAN 1-km grid and the 1-min results from the WRF 1.5-km grid are then used to force the unstructured mesh of the ADCIRC-SWAN model. In this deterministic configuration, the ADCIRC model is forced every minute by the WRF 1.5-km wind and pressure fields, and every hour by the basic module sea-level fields (including tides) at the open sea boundary (Otranto Strait)."

Second bullet point is addressed with the following paragraph:

"The observational network (called MESSI, www.izor.hr/messi) consists of nine microbarographs, of which eight are used in this study, measuring air pressure by the Väisälä PTB330 sensor with an accuracy of ±0.01 hPa, and three tide gauges, of which two are used in this study, measuring sea-level by the OTT RLS radar level sensor with an accuracy of ±1 mm. All instruments are setup with a 1-min sampling rate and listed in Table 1."

The last bullet point is already addressed in the "Methods" section defined above.

*line 75: "(2) measurements from the MESSI (www.izor.hr/messi) observational network" does not provide useful information. I suggest replacing with: "(2) high-frequency air pressure and sea level measurements along ..."*

**Response:** Accepted. We replaced it with:

"(2) high-frequency air pressure and sea-level measurements along the Adriatic coast…"

*line 84: ... with the 2D unstructured ADvanced CIRCulation (ADCIRC) model ...*

**Response:** Accepted. We replaced with:

"with the 2DDI (i.e. two dimensional depth-integrated) unstructured ADvanced CIRCulation (ADCIRC) model"

*line 87: ... sea-level fields (including tides) at the open sea boundary (Otranto Strait).*

**Response:** Accepted. We replaced the existing line with:

"…sea-level fields (including tides) at the open sea boundary (Otranto Strait)."

*I strongly suggest splitting Figure 1 in two: the first containing only the map and putting all time series on a separate figure 2. Depth should be positive.*

**Response:** In order to limit the number of figures in the article (which is already large), we kept the figure as one but have put all the time series in two columns below the map, to fit with the journal format. Also, we changed the depth sign into positive.

*Remove lines 98-101.*

**Response:** Accepted. Those lines are removed.

*line 110: The observational network (called MESSI, www.izor.hr/messi), ...*

**Response:** Accepted. We replaced the existing line with:

"The observational network (called MESSI, www.izor.hr/messi)..."

*line 116: ... at the tops of the bays that are normally most affected ...*

**Response:** Accepted. We changed the existing line to:

"…are located not at the tops of the bays that are normally most affected by meteotsunamis, but about 2 km from the tops."

*Figure 3 should be moved below in the text.*

**Response:** Accepted. The figure is moved below in the text.

*Page 10: I suggest to use the full name of the monitoring stations instead of their abbreviation.*

**Response:** Accepted. Full names of the monitoring stations are now used instead of abbreviations.

*Lines 274-298: this part should be moved before (or at the beginning) of section 4.*

**Response:** Accepted. This part is now a part of a new methodology section introduced after the introduction section.

*Figures 6 to 10: please include labels for the transect's beginning and end (e.g. A and B) in maps and spectrograms or specify that all transects are plotted from the west to the east. It is unclear what's the transect number.*

**Response:** Accepted. It is added that all transects are extracted from West to East.

*Lines 425-427 and 435-438: In both sentences, it's written that the ocean model fails in predicting meteotsunamis. As it is written it seems that the problem resides in the ocean model itself, while most of the uncertainty is associated with the atmospheric modelling of the meteotsunamigenic disturbance, as written in the subsequent phrases. I suggest reformulating the text in order to clarify that without accurate atmospheric predictions there are no chances to forecast meteotsunamis.*

**Response:** Accepted. "due to a shift in location of the modelled atmospheric disturbances" is added to both sentences.

**Anonymous Reviewer #2 (RC2)**

*The present paper deals with the forecasting of meteotsunamis in some ports of the Croatian Coast, one of the world areas where this kind of phenomena acquire relevant magnitude. Forecasting of meteotsunamis (that is, large sea level oscillations in a range of periods similar to the tsunami periods, triggered by small scale meteorological disturbances) is an operational and scientific*

*challenge, due to the complexity of the phenomenon and the small scale of the meteorological perturbations directly triggering the meteotsunamis. The Croatian scientists have implemented a meteotsunami forecasting/warning system (CMeEWS) constituted by a numerical model prediction suite (including atmospheric and oceanic modules), an observational meteorological and oceanic system and a stochastic surrogate model.*

*The particular objective of this paper is to check the behaviour of CMeEWS during a recent large period of meteotsunamis in some of the Croatian ports (11 to 19 May, 2020). As a result, the authors are not optimistic with regard to the deterministic forecasts directly obtained from the numerical model suite and they rely more in the probabilistic forecasts obtained from the stochastic model. The authors highlight the introduction of a verification method, based on the examination of the energy banners associated to the displacement of meteorological small scale perturbations, as one of the main results of their work.*

**Response:** Thank you very much for all your comments. The article is indeed dedicated to the evaluation of the CMeEWS during the 2020 multi-meteotsunami event using energy banners. It must be said that both the observational network and the modelling suite implementations in the Adriatic Sea are extremely recent. It is consequently crucial to better understand the performances of such a system and to gain knowledge on the quality of the provided hazard assessments.

*In fact, one of the general concerns of this referee with regard to this paper is that the authors seem not to be very clear in their objectives and results. An effort to review the text in this sense could improve the paper.*

*This is an aspect of a more general problem of this paper: it is quite complicate in its drawing, what makes a little difficult to read it. A general review of the drawing is convenient.*

**Response:** The authors agree with the reviewer and have introduced a new "Methods" section in order to clarify the objectives of each analysis done in the article. Hopefully, this will help readers follow the work with ease.

*The high degree of uncertainty in the deterministic forecasting of some small scale meteorological disturbances, particularly those related with the triggering of meteotsunamis, is a well-known fact, although there are cases in which particular disturbances of this kind can be reasonably well forecasted is rare (Renault et al, 2011).*

*To reduce or to narrow down the uncertainty of the deterministic forecast, CMeEWS includes a stochastic model based on the polynomial chaos expansion method. The stochastic model provides probabilistic forecasts that the authors consider more useful than the direct deterministic forecasts. The model used in CMeEWS is a way, but not the only way to narrow down the uncertainties though probabilistic forecasts: Vich and Romero (2020: https://doi.org/10.1007/s11069-020-04041-5) or Mourre et al (2020: https://doi.org/10.1007/s11069-020-03908-x)*

**Response:** The authors agree with the reviewer that the uncertainty linked to the meteotsunami forecast is high and this is why they developed their own methodology to solve this issue. The suggested references with methodologies that aims to narrow down uncertainty of deterministic forecasts are added in the introduction as follow: "In the Balearic Islands, probabilistic approaches have also been tested recently to narrow down the uncertainties of the meteotsunami forecasts (Vich and Romero, 2020; Mourre et al., 2020)"

*Regarding Section 2, although this paper is not the presentation of the CMeEWS (this was made in previous papers), this system is also described here, although not with enough clarity. It seems to me that the authors describe a numerical prediction model suite, which is a part of the CMeEWS and that contains a basic module, named COASTWST, and a meteotsunami module. I understand that the meteotsunami module includes the known atmospheric model WRF, running at a resolution of 1-1.5 km, and providing air pressure data, every minute, to a marine module (ADCIRC), which resolutions is variable, reaching up to 100 m in the most sensible zones. Is that correct? Other key details are not explained. Particularly, which are the models that constitute the basic module COASTWST? Which are their characteristics? How they feed WRF, as part of the meteotsunami module?*

**Response:** The authors agree with the reviewer that the modelling suite was not described in detail in this particular article. The following extension of the subsection describing the AdriSC modelling suite is added in the text:

"The basic module uses a modified version of the Coupled Ocean-Atmosphere-Wave-Sediment-Transport (COAWST) modelling system developed by Warner et al. (2010). The system is built around the Model Coupling Toolkit (MCT) which exchanges data fields and dynamically couples the Weather Research and Forecasting (WRF) atmospheric model, the Regional Ocean Modeling System (ROMS), and the Simulating WAves Nearshore (SWAN) model. The basic module is set-up with (1) two different nested grids of 15-km and 3-km resolution used in the WRF model and covering respectively the central Mediterranean area and the Adriatic-Ionian region and (2) two different nested grids of 3-km and 1-km resolution used for both ROMS and SWAN models and covering respectively the Adriatic-Ionian region (similarly to the WRF 3-km grid) and the Adriatic Sea only.

The dedicated meteotsunami module couples offline the Weather Research and Forecasting (WRF) model (Skamarock et al., 2005) with the 2DDI (i.e. two-dimensional depth-integrated) unstructured ADvanced CIRCulation (ADCIRC) model (Luettich et al., 1991), using a mesh of up to 10-m in resolution in the areas sensitive to meteotsunami hazard. In more details, (1) the hourly results from the WRF 3-km grid obtained with the basic module are first downscaled to a WRF 1.5-km grid covering the Adriatic Sea and (2) the hourly sea surface elevation from the ROMS 1-km grid, the 10-min spectral wave results from the SWAN 1-km grid and the 1-min results from the WRF 1.5-km grid are then used to force the unstructured mesh of the ADCIRC-SWAN model. In this deterministic configuration, the ADCIRC model is forced every minute by the WRF 1.5-

km wind and pressure fields, and every hour by the basic module sea-level fields (including tides) at the open sea boundary (Otranto Strait)."

*Section 2.3, line 126, I don't understand what "fail to reproduce or underestimate" means.*

**Response:** The statement that deterministic ocean models often fail to reproduce or underestimate the meteotsunami events in sensitive harbours means that sea-level oscillations in those harbours is either non-existing ("fail to reproduce" case) or have an intensity significantly lower than the real ones ("underestimation" case).

*Lines 127-128, please, clarify the sentence*

**Response:** Accepted. The sentence is changed to:

"In order to improve the meteotsunami hazard assessments in the Adriatic, the meteotsunami stochastic surrogate model, used to propagate the uncertainties of the atmospheric disturbance parameters extracted from the WRF 1.5-km model to the maximum amplitudes of the meteotsunami waves, was developed within the CMeEWS (Denamiel et al., 2019b, 2020)."

*Line 135, how the parameters of the atmospheric perturbation are obtained from WRF? automatically? subjectively?*

**Response:** The parameters of the atmospheric perturbation are extracted manually by analyzing the results of the air pressure signal. This is added to the text as follow:

"Within the CMeEWS, the ranges of the stochastic parameters used as input to the surrogate model are extracted manually from the forecasted WRF 1.5-km high-pass filtered air pressure results, adding the uncertainty of $\pm 0.24°$ N for latitude of origin, $\pm 0.26$ rad for direction of propagation, $\pm 0.35$ hPa for amplitude, $\pm 150$ s for period, and $\pm 12$ km for width, following the values determined by Denamiel et al. (2019b)."

*Section 3, figure 2: this figure is not clear. It is difficult to see the lines with clarity. Perhaps it would be better to remove the colours. On the other hand, perhaps adding wind in the 850 hPa panels would help to the meteorological large scale frame. Another suggestion: perhaps including a vertical atmospheric profile would also help.*

**Response:** The figure is a classical presentation of synoptic conditions that are documented in a great number of papers to conjoin meteotsunamis: (1) inflow of warm air at 850 hPa, (2) winds (speed and direction) at 500 hPa, and (3) mean sea-level pressure. So, adding winds at 850 hPa will just blur the temperature behind. Also, removing colors will make the figure less readable, while colours represent standard choices as it stands now – red denotes a warm air and also high wind speed (or mid-troposhere jet in our case).

*Section 4, figures 3, 4, 5. What are the abscises in every box? Are they time? Following lines 214-215 it seems that they are time and, particularly, twelve hours around the time of interest: is it correct? If so, the times are not indicated. How have the places been selected? Is "composite" referred to spectral analyses over 30 minutes sampling? Running, continuous overlapping or discrete sampling?*

**Response:** The authors agree that the figures may have been slightly confusing. The abscises are time, 24h for each day indicated on the bottom of the figure. Time is added on the figures. These places were selected because of the available microbarograph data at these locations. Selection of model grid points W1-W7, M1-M2 and E1-E6 and the composite term is explained in the new "Methods" section as follow:

"Since the failure of deterministic models to reproduce the small scale atmospheric disturbances at the right locations is a well-known problem, the verification of the AdriSC WRF 1.5-km results presented in Section 5 tracks the locations where the highest daily spectral energies occur in both the model and the observations. In other words, the performance of the AdriSC WRF 1.5-km model is derived with Fast Fourier Transforms (FFT) analyses (Cooley and Tukey, 1965) of the high-pass filtered air pressure observed and modelled results calculated every 30 min with a 3-h window at selected locations for each day of the reproduced multi-meteotsunami event. First, as the meteotsunamigenic disturbances are known to propagate from the Western to the Eastern Adriatic (Vilibić and Šepić, 2009; Denamiel et al., 2020), 5 transects are selected to track the modelled atmospheric disturbances: 2 transects along the Italian coast in the Western Adriatic (T4 and T5), one in the Middle Adriatic (T3) and two transects along the Croatian coast in the Eastern Adriatic (T1 and T2). Then, for each day of the multi-meteotsunami event, the AdriSC WRF 1.5-km results are extracted at the actual microbarograph locations and in additional model grid points (black dots, Fig.1) selected where the highest daily spectral energies are reproduced by the model along the Western (selected points W1 to W7), Middle (selected points M1 and M2), and Eastern Adriatic (selected points E1 to E6) transects. Finally, for each day with a meteotsunami event, the time evolutions of the spectra derived from the observations (at the microbarograph location where the meteotsunami was best observed – i.e. highest spectral energy along the Western Adriatic transect for Ancona, Ortona and Vieste microbarographs, along the Middle Adriatic transect for Vis and Svetac microbarographs and along the Eastern Adriatic transect for Vrboska, Stari Grad and Vela Luka microbarographs) are compared with the time evolutions of the spectra derived from the WRF 1.5-km results at the point where the highest energy was reproduced (including microbarograph locations). At the end, for the entire duration of the multi-meteotsunami event, composites of frequency-time spectrograms of high-pass filtered air pressure observed and modelled data for the Western, Middle and Eastern Adriatic regions are created (Figs. 3-5)."

*lines 204-206, no a threshold for air pressure change is mentioned. Is it 20 Pa/4 min, as indicated by Denamiel et al (209b)? Please, review drawing*

**Response:** Accepted. Yes, the threshold is 20 Pa/4 min, as indicated by Denamiel et al (2019b). The following paragraph is also added to the "Methods" section:

"Within the CMeEWS, the meteotsunamigenic disturbances reproduced with the AdriSC WRF 1.5-km model are automatically detected if the maximum temporal rate of change (i.e. pressure difference calculated over a 4-min interval) of the high-pass filtered air pressure derived at each WRF 1.5-km grid sea point is above 20 Pa/min over at least 5% of the sea domain. Such a condition has been proven to be efficient for the detection of meteotsunamigenic disturbances (Vilibić et al., 2016; Denamiel et al., 2019b). The event mode of the system (i.e. meteotsunamis may occur) is thus triggered without human interventions for the studied 11-19 May 2020 period."

*Line 206, "are greater than" (??) à"is greater than"*

**Response:** Accepted. The sentence is moved to the "Methods" section and reformulated as shown in the above response.

*Line 216, "intense air pressure" is an incomplete or not understable expression*

**Response:** Accepted. This part of the sentence is changed to "…which is typical for spectra of air pressure characterized by a number of oscillatory movements with no dominant period…"

*Lines 217-220: the authors indicate the System renounces to indicate a timing for the phenomenon. It is pity don't include indication about the timing: it would be a potential added value to the daily warnings*

**Response:** We agree with the reviewer that discussing the timing of the meteotsunami events is definitely important and would add to the hazard assessments provided by the CMeEWS. However, at the present, the system has already to be proven reliable in terms of capturing the meteotsunami events for a 24-h period before trying to predict the timing at which they will occur (which is even more complicated and uncertain). In any case, some efforts are under way to create a surrogate model for the time.

*Section 5, line 281, is it spatial variance (in an area) at a fixed time or time variance (during 3 hours) at fixed grid points? It seems the second. Please, review*

*Line 283, it seems that the points of largest variance define energy banners or transects. Are these transects determined objectively or subjectively, transect by transect?*

*Lines 293-294, I don't understand well the definition of "transect sampling criteria", is it a magnitude obtained multiplying air pressure variance and marine response? Please explain a little more*

**Response:** All above three comments are accepted. We added the following paragraph in the methodology section.

"The analyses performed in Section 6 are done in two steps and aim to better track the propagation of the modelled meteotsunamigenic disturbances across the Adriatic Sea, in order to improve the extraction of the atmospheric parameters needed to run the stochastic surrogate model. In the first step, two different transect sampling criteria are used to select the transects along which the atmospheric disturbances, and hence the meteotsunami waves, propagate in the model: one based solely on the atmospheric results (already used operationally) and a new one also taking into account the ocean results (tested in this study). For the operational sampling criterion, the time variances of the WRF 1.5-km high-pass filtered air pressure results are calculated on a 3-hour interval (i.e. 8 time-windows per day) over the entire model domain. For each event occurring during the 11-19 May 2020 period, the transects presented in this study are selected across the Adriatic Sea following the paths of highest atmospheric variances for the most energetic time-windows. Since the number of time-windows and paths with high air pressure variances varies between the events, the number of transects for each day varies too. For the new sampling criterion, the variances of the high-pass filtered air pressure and sea-level model results estimated on a 3-hour interval are multiplied. This criterion thus tends to zero when the atmospheric forcing does not trigger any ocean response, i.e. when no resonant transfer of energy from the atmosphere to the sea is occurring. It should be noted that such a criterion could not be directly derived from the sea-level variances which provide a noisy and mostly untraceable signal due to the numerous interactions of the ocean waves with the bathymetry including, for example, the reflection and refraction around the islands. Hereafter the new transect sampling criterion is compared with the operational one in order to determine whether or not it would have improved the transect selection."

*Figures 6 to 10 and the corresponding figure captions. Some questions: How the speed of propagation of the pressure disturbances is determined? In the down panels of these figures it seems that the abscises are distances along the transect, but no scale is indicated. No all the possible transects are explicitly considered in these figures; in the supplement, figures 2 to 15 are more exhaustive It is no clear how the transects are chosen for figures 6 to 10. The reference to the figures of the supplement in the captions of figures 6 to 10 is more confusing than clarifying. In my opinion, all the captions, from figure 6 to 10 have to be written more clearly. Some particular details:*

**Response:** Accepted. In the "Methods" section, the methodology concerning the extraction of the atmospheric disturbance speeds is added as well as the reason why not all transects were presented within the main article.

"In the second step, meteotsunami energy banners defined as the spectrograms of the modelled high-pass filtered air pressure and sea-level results are spatially calculated with FFT along the selected transects for the 3-h time-window corresponding to the operational transect sampling criterion. As speed remains a difficult parameter to extract from the observed and modelled

meteotsunamigenic disturbances, speeds of the tracked atmospheric disturbances along the transects are also visually determined by analysing the propagation along the transects of the strongest WRF 1.5-km high-pass filtered air pressure peaks. The locations where the Proudman resonance is likely to occur along the transects are then derived by calculating where the Froude number ($Fr=U/C$) ranges from 0.9 and 1.1 (i.e. where the speed of the atmospheric disturbances $U$ are matching the speed of the long ocean waves $C=\sqrt{gH}$, with $g$ the gravitational acceleration and $H$ the local depth). The analyses from Section 6 are presented with one transect (plotted from West to East following the propagation of the meteotsunami events) per event in the article (Transect 1, Figs. 6-10) selected during the peak of the modelled daily event and as supplementary material for the other transects (Figs. S2-S15) in order to keep a reasonable article length."

Additionally, the reference to supplementary material in the captions of Figures 6 to 10 are removed and the scale of the abscises is added.

*Figure 7: With regard to the transect that is highlighted, it is not easy to understand why the transect sampling criteria –second panel in the figure- give so large values, when there are not condition for a Proudman resonance –last panel-.*

**Response:** We agree that this is interesting result that might be related to the fact the Proudman resonance is not the only process responsible to generate meteotsunami waves in the middle Adriatic, where the bathymetry is changing rapidly. Šepić et al. (2016) highlighted that "over complex bathymetries, like the middle and south Adriatic ones, numerous effects, including Proudman resonance, edge waves, strong topographical enhancement and refractions on the islands placed on the pathway of atmospheric disturbances should be taken into account to fully understand meteotsunami generation and dynamics".

We expanded the text when introducing Fig. 7 as follows:

"Nevertheless, the transect is in a deep water with changing bathymetry, and therefore the Proudman resonance is only likely to happen over a small part of the transect, while other effects, including edge waves, strong topographical enhancement and refractions on the islands placed on the pathway of atmospheric disturbances may be important for generation of meteotsunami waves in the middle Adriatic (Šepić et al., 2016)."

We also added the cited reference to the reference list:

Šepić, J., Međugorac, I., Janeković, I., Dunić, N., and Vilibić, I.: Multi-meteotsunami event in the Adriatic Sea generated by atmospheric disturbances of 25–26 June 2014. Pure and Applied Geophysics, 173, 4117-4138, https://doi.org/10.1007/s00024-016-1249-4, 2016.

*Figure 8: The caption mentions two transects, but only one is indicated (on the contrary, in figure 6 the caption only mentions one transect, but the first panel seems to show two transects).*

**Response:** The other transects are presented in the supplementary material in Figures S2, S3 and S7.

*Figure 9: Only one transect is indicated in the first panel, but it seems to me that there are two parallel partial transects, one of them vanishing soon, the other appearing late*

**Response:** Accepted, we agree with the reviewer. There are more than one transect that can be observed in Figure 9, these transects are presented in the supplementary material in Figures S8 and S9.

*Lines 381-383: Difficult to understand; please, clarify*

**Response:** Accepted. The clarification of the statement is added in the following form:

"Finally, the introduced new transect sampling criterion does not seem to overall facilitate the decision-making process in terms of the transect selection, since all the transects selected by this criterion would have also been selected following highest values of air pressure variances only. Even though for some events (e.g. Figs. 8, 9, 10) the new criterion highlights the strength of the air-sea interactions, these interactions are located along the same transects as captured by the highest values of the air pressure variance. As efficiency is important in early warning system, it can thus be concluded that the use of the ocean model results to better select the transect with maximum meteotsunami generation is not necessary in operational mode, since it would be more time consuming with no significant value added to the process of the transects selection.".

---

## Author Response (AR2)

**Response to the reviewers' comments on the manuscript "Performance of the Adriatic early warning system during the multi-meteotsunami event of 11-19 May 2020: an assessment using energy banners" by Tojčić et al., submitted to NHESS (nhess-2020-409)**

The authors, despite being extremely surprised to discover major revisions in the second review that were not mentioned in the first one, appreciate the careful consideration given by the reviewer to the presented work. Most of the revisions were implemented in the new version of the article and they definitely improve the manuscript. However, the authors feel that major revisions asked in points 3 and 5 are essentially based on a misunderstanding of what early warning systems can achieve. First, the presented CMeEWS is a research product still under development and, second, even well-funded early warning systems, such as the NOAA hurricane early warning system (NHC, https://www.nhc.noaa.gov) or the CEA European tsunami early warning system (CENALT, Schindele et al., 2015), often fail to forecast extreme sea-levels and require many human interventions. The authors hope their detailed arguments will suffice to convince the reviewer of the quality of the presented methods and results.

**Anonymous Reviewer #1**

*The description of the "The Croatian Meteotsunami Early Warning System" (section 2) should be placed inside the Methods section.*

**Response:** Accepted. The description of the Croatian Meteotsunami Early Warning System is moved to the now renamed Model, Data and Methods section.

*The observation/model comparison presented in the bottom panels of Fig. 1 is too small and therefore not useful for model validation (even qualitatively). Therefore, I strongly suggest putting these panels in a separate figure with appropriate labels.*

**Response:** Accepted. The panels are put in a separate figure (Figure 2).

*In my opinion, the use of the energy banners presented in figures 3-5 and discussed in Section 5 could not be used for providing a quantitative assessment of the model performance. I do understand that simulating meteotsunamis and their tsunamigenic conditions at the right timing and location is challenging and therefore it is correct to analyze model results in the neighbourhood of the monitoring station. However, it is not acceptable - even considering the scale of the perturbations - to present a comparison using model results extracted at locations hundreds of kilometres far from the monitoring stations (e.g. Or-W2, Ve-W1, .. in Fig. 3). Therefore, figures 3-5 are mostly useless for the model assessment. The authors should consider only results at a reasonable distance from the monitoring station.*

**Response:** The authors believe that the above comment of the reviewer is linked to an inherent misconception of the methodology proposed in Section 4 (previously Section 5).

First, as a reminder, the methodology is comparing - along the western coast, the middle and the eastern coast of the Adriatic - the time evolution of the spectral analysis of the strongest atmospheric disturbances modelled and measured.

Second and foremost, the observation network is extremely sparse as, for example, along the Italian coast only three stations cover around 600-km of coastline. It is thus impossible to know if the recorded meteotsunamigenic disturbances at the Ancona, Ortona and Vieste stations are the maximal disturbances during the meteotsunami event. Indeed, these stations are separated by 150-km to 200-km of coastline and the "real" strongest meteotsunamigenic disturbances can occur anywhere between the stations. In this sense, the fact that the strongest model results are extracted 100-km further from the station which recorded the strongest meteotsunamigenic disturbances is not necessarily pointing to the location of the atmospheric disturbance in the model being 100-km further from the strongest real disturbance. Obviously, the comparison Ve-W1 mentioned by the reviewer is typically an excellent example when the model is definitely generating disturbances far too north compared to the reality. This leads to our third point.

Third, "Essentially, all models are wrong, but some are useful." (George E. P. Box, Robustness in the strategy of scientific model building, 1979). In this article, the authors are showing the usefulness of the deterministic component of the AdriSC modelling suite which provides atmospheric parameters to the stochastic component despite being sometime/often "essentially wrong". It is worth mentioning here that the operational model forecasting meteotsunami in the Balearic Islands (to this date, the only meteotsunami forecast running continuously for years) also struggles to deterministically predict meteotsunamis in Menorca (Mourre et al., 2020). As mentioned in Denamiel et al. (2019), the deterministic forecast of meteotsunami requires to push the use of the atmospheric state-of-the-art models beyond their original goals. As a consequence, it is not surprising that deterministic forecast of meteotsunamis often fail.

In brief, Section 4 provides a first estimate on how the atmospheric forecast of the AdriSC modelling suite succeeds to capture the intensity and presence but fails to reproduce the location of the observed meteotsunamigenic disturbances. It is in this sense an honest evaluation of the forecast capacity compared to the sparse observational network available in the Adriatic Sea. Additionally, Section 4 demonstrates the absolute necessity to use the stochastic component of the AdriSC modelling suite.

*Even if no tsunamigenic disturbances were recorded during the 12th and 13th May, it would be useful to see model results for this period, also for checking if the system provides false alarms.*

**Response:** Accepted. The model has been run for 12[th] and 13[th] of May. Results are presented in Figures 2, 4, 5 and 6, and analyzed in Section 4.

*The aim of this study is to quantify the performance of the Croatian meteotsunami early warning system (CmeEWS). Such a system has been run retroactively in operational (hindcast) mode. It is however not clear to me if potentially the system could be used in operational mode since many operations depend on the direct human interventions (e.g. selection of transects and extraction of the input parameters of the stochastic surrogate model). The authors should provide a clear scheme of the operational setup including the role of forecast operators.*

**Response:**

As explained in the introduction of the article: "… the recently developed Croatian Meteotsunami Early Warning System (CMeEWS) is based on an observational network of pressure sensors and tide gauges, as well as on the deterministic AdriSC modelling suite (Denamiel et al., 2019a) and the stochastic meteotsunami surrogate model (Denamiel et al., 2019b, 2020). It provides meteotsunami hazard assessments depending on forecasted and measured air pressure disturbances but is, unfortunately, not used operationally since November 2019 due to a lack of high-performance computing resources needed to execute in real-time such numerically demanding suite."

The CMeEWS has thus been run in operational mode for about a year after which, due to a lack of sustainable funding and available numerical resources, it was unfortunately stopped. The authors hope to re-start the operational system in a near future and, meanwhile, decided to continuously develop/evaluate the numerical models with every new meteotsunami event.

[Figure]

Figure R1. Extracted from Denamiel et al. (2019a)

The scheme of the operational setup was already provided in previous studies (Denamiel et al. 2019a, 2019b) and are presented in Figures R1 and R2.

[Figure]

Figure R2. Extracted from Denamiel et al. (2019b)

As the reviewer can see, the Manual Extraction step is clearly defined within the CMeEWS (Figure R2). This is obviously when the intervention of the forecast operators is required.

It should be also pointed out that human interventions are often a pre-requisite of early warning systems. For example, in CEA, where the European tsunami early warning system (CENALT) is based, operators are constantly (24/7) monitoring the observational system and their roles are (1) to validate the automatic treatment of the data, (2) to eventually correct them if necessary and (3) to run the software dedicated to the computation of the timing and location the extreme sea-levels along the Mediterranean coastline generated by the recorded offshore tsunami waves (http://www.info-tsunami.fr/content.php?sec=27).

Consequently, the authors believe that the point raised by the reviewer has already largely been answered in the previous studies dealing with the CMeEWS, which is in fact following the most common practices of human intervention in early warning systems.

*Lines 483-485: this is just a speculation, not the result coming from analysis. Please, remove this sentence or provide a detailed analysis supporting this statement.*

**Response:** Accepted. The sentence is removed from the manuscript.

*line 107: the unstructured currect-wave model ADCIRC-SWAN is here mentioned but SWAN is described only as part of the COAWST system.*

**Response:** The description of the Nearshore module of the AdriSC modelling suite has been amended and now reads: "The dedicated meteotsunami module couples offline the Weather

Research and Forecasting (WRF) model (Skamarock et al., 2005) at 1.5-km of resolution with the unstructured ADCIRC-SWAN model (Dietrich et al., 2012) coupling the 2DDI (i.e. two dimensional depth-integrated) ADvanced CIRCulation (ADCIRC) model and the SWAN model with a mesh of up to 10-m resolution in the areas sensitive to meteotsunami hazard."

*line 203: how are the transects selected? Manually?*

**Response:** Yes, the transects were selected manually. This is added in the text: "For each event occurring during the 11-19 May 2020 period, the transects presented in this study are manually selected across the Adriatic Sea following the paths of highest atmospheric variances for the most energetic time-windows."

*line 215: what do you mean by "visually determined"?*

**Response:** Visually determined means that the analysis of the plots of filtered air pressure along the transect was done. The distances over which a peak of the disturbance travelled in a certain period of time were determined from these plots. The speed was then easily obtained from distance and time values.

*line 283-284: not proven, it would be better to skip the sentence.*

**Response:** Accepted. The sentence is removed from the manuscript.
* * *
**Reference:**

Schindelé, F., Gailler, A., Hébert, H. et al.: Implementation and Challenges of the Tsunami Warning System in the Western Mediterranean. Pure Appl. Geophys. 172, 821–833 (2015). https://doi.org/10.1007/s00024-014-0950-4